



# Modelling snowpack dynamics and surface energy budget in boreal and subarctic peatlands and forests

Jari-Pekka Nousu[1,2,3], Matthieu Lafaysse[2], Giulia Mazzotti[2], Pertti Ala-aho[1], Hannu Marttila[1], Bertrand Cluzet[4], Mika Aurela[5], Annalea Lohila[5,6], Pasi Kolari[6], Aaron Boone[7], Mathieu Fructus[2], and Samuli Launiainen[3]

[1]Water, Energy and Environmental Engineering Research Unit, University of Oulu, Oulu, Finland
[2]Univ. Grenoble Alpes, Université de Toulouse, Météo-France, CNRS, CNRM, Centre d'Études de la Neige, Grenoble, France
[3]Bioeconomy and Environment, Natural Resources Institute Finland, Helsinki, Finland
[4]WSL Institute for Snow and Avalanche Research SLF, Davos, Switzerland
[5]Climate System Research, Finnish Meteorological Institute, Helsinki, Finland
[6]Institute for Atmospheric and Earth System Research INAR, University of Helsinki, Helsinki, Finland
[7]CNRM, Université de Toulouse, Météo-France, CNRS, Toulouse, France

**Correspondence:** Jari-Pekka Nousu (jari-pekka.nousu@oulu.fi)

**Abstract.** The snowpack has a major influence on the land surface energy budget. Accurate simulation of the snowpack energy budget is challenging due to e.g. vegetation and topography that complicate the radiation budget, and limitations in theoretical understanding of turbulent transfer in the stable boundary layer. Studies that evaluate snow, hydrology and land surface models (LSMs) against detailed observations of all surface energy components at high latitudes are scarce. In this study, we compared

different configurations of SURFEX LSM model against surface energy flux, snow depth and soil temperature observations from four eddy covariance stations in Finland. The sites cover two different climate and snow conditions, representing the southern and northern subarctic zones, and the contrasting forest and peatland ecosystems typical for the boreal landscape. We tested the sensitivity of surface energy fluxes to different process parameterizations implemented in the Crocus snowpack model. In addition, we examined common alternative approaches to conceptualize soil and vegetation, and assess their perfor-

mance in simulating surface energy fluxes, snow conditions and soil thermal regime. Our results show that using a stability correction function that increases the turbulent exchange under stable atmospheric conditions is imperative to simulate sensible and latent heat fluxes over snow. For accurate simulations of surface heat fluxes and snow/soil conditions in forests, an explicit vegetation representation is necessary. Moreover, we found the peat soil temperature profile simulations to be greatly improved with realistic soil texture (soil organic carbon) parameterization. Although we focused on models within the SUR-

FEX LSM platform, the results have broader implications for choosing suitable turbulent flux parameterization and model structures depending on the potential use cases.



# 1 Introduction

The boreal zone, characterized by a mosaic of seasonally snow-covered peatlands, forests and lakes, is the largest land biome in
the world. Snow conditions in the boreal zone are rapidly changing due to climate warming, which is found to be the strongest
during the cold seasons in the Arctic (Serreze et al., 2009; Screen and Simmonds, 2010; Boisvert and Stroeve, 2015; Ranta-
nen et al., 2022). Evidently snow has an important role for water resources and human activities in the cold regions, but it
is also known that the snowpack characteristics affect animal movement (Tyler, 2010; Pedersen et al., 2021) and plant distri-
bution (Rasmus et al., 2011; Kreyling et al., 2012; Rissanen et al., 2021). Recent studies show that especially cold-dwelling
species have been shifting towards higher latitudes and altitudes in search for more suitable habitats (Tayleur et al., 2016; Couet
et al., 2022). Therefore, the rapid warming of the Arctic, and its consequences on the quantity and properties of snow may de-
fine the destiny of many species and human activities in the boreal region. To predict future snow conditions, environmental
change, and the consequences for water resources, ecosystems and people, predictive and process-based models possess great
potential (Clark et al., 2015; Boone et al., 2017). Land surface models (LSMs) have been used for decades in numerical weather
prediction (NWP), global circulation models (GCMs) (Douville et al., 1995; Niu et al., 2011; Lawrence et al., 2019), and have
more recently become common tools for interdisciplinary impact studies (Blyth et al., 2021).

Snowpack has a major impact on the wintertime energy budget due to its influence on the land surface broadband albedo
(LSA) and the surface heat fluxes (Cohen and Rind, 1991; Eugster et al., 2000). The heat diffusion within the snowpack is
determined by the surface heat fluxes, internal properties of the snowpack and soil thermal regime. Correctly representing the
snowpack is thus essential for simulating energy and mass exchange between the snow surface and the atmosphere, as well
as below the snowpack (e.g. surface temperatures and soil freezing/thawing dynamics, Koivusalo and Heikinheimo, 1999;
Slater et al., 2001). The snowpack energy budget is partitioned into downwards and upwards shortwave (SWD, SWU) and
longwave (LWD, LWU) radiation, turbulent fluxes of sensible (H) and latent heat (LE), snowpack-ground heat flux (G) and
phase-changes in the snow. The snowpack energy balance and energy partitioning among the flux components vary strongly
across diurnal and seasonal timescales, and between different ecosystems (Clark et al., 2011; Stiegler et al., 2016; Stigter et al.,
2021). It is essential that LSMs are able to correctly reproduce this variability.

On the vast boreal and arctic peatlands with shallow vegetation, the snow cover can exclusively determine the wintertime
LSA (Aurela et al., 2015). With minimal solar radiation during winter months on these open snow fields, turbulent fluxes make
an important component in the energy budget of the snowpack, as they compensate the radiative cooling processes and further
contribute to snow melt (Lackner et al., 2021; Conway et al., 2018). Simulation of turbulent fluxes under stable atmospheric
conditions is known as one of the major sources of uncertainty in snow models (Lafaysse et al., 2017; Menard et al., 2021). In
LSMs the turbulent fluxes are commonly computed with bulk aerodynamic approaches, where H and LE are proportional to the
turbulent exchange coefficient according to the Monin-Obukhnow similarity theory (MOST). These approaches typically use
atmospheric stability correction functions based either on the bulk Richardson number (Martin and Lejeune, 1998; Lafaysse





et al., 2017; Clark et al., 2015) or the Obukhov length scale (Jordan et al., 1999). It is established that MOST theory does not
well represent low-wind and stable atmospheric conditions above aerodynamically smooth surfaces such as snow (Conway
et al., 2018). In such conditions, the simulated surface temperatures have been found to be unrealistically low, as the turbulent
boundary layer tends to decouple from the snow surface (Derbyshire, 1999; Andreas et al., 2010). To circumvent this effect,
stability correction functions have been modified to permit turbulent fluxes above critical stability thresholds (Lafaysse et al.,
2017), by manipulating the wind speed (Martin and Lejeune, 1998; Andreas et al., 2010), or including a windless turbulent
exchange coefficient (Jordan et al., 1999). Evaluations of these modifications often rely on validation with observed surface
temperatures and snow depths (e.g. for the detailed snowpack model Crocus, Martin and Lejeune, 1998; Lafaysse et al., 2017)
while comparisons against turbulent energy flux data remain scarce (Lapo et al., 2019; Conway et al., 2018).

The energy budgets of forest canopies and below-canopy snowpack are different to those on open peatlands, as turbulent
exchange is attenuated by the canopy, and the snowpack energy budget and snow melt are mostly driven by the radiation
balance (Rutter et al., 2009; Essery et al., 2009; Varhola et al., 2010). However, due to heterogeneous canopy structures and
canopy processes (radiation transmittance, snow interception and unloading) together with low solar angles, the dynamics of
LSA in seasonally snow-covered boreal forests is complex (Malle et al., 2021). The absorption of the shortwave radiation can
be highly heterogeneous in forest stands, having direct implications on canopy temperatures (Webster et al., 2017), and on the
resulting longwave radiative fluxes between canopy, snowpack and the atmosphere (Mazzotti et al., 2020b). Forest snow mod-
elling has been identified as a priority in advancing cold region climate and hydrological models (Rutter et al., 2009; Krinner
et al., 2018; Lundquist et al., 2021). Various models that have been proposed to represent the large scale impact of forest on the
snowpack energy budget (Niu et al., 2011; Lawrence et al., 2019; Boone et al., 2017) are still prone to large errors, due to the
complexity and unresolved spatial scales of the underlying physical processes (Loranty et al., 2014; Thackeray et al., 2019).
The forest snow model evaluations against concurrent snowpack and surface energy balance data are also surprisingly scarce.
For instance, the explicit forest scheme of SURFEX LSM, MEB (Multi-Energy Balance, Boone et al., 2017) has so far been
evaluated only against data from three neighbour sites in Saskatchewan, Canada (Napoly et al., 2020). This considerably limits
knowledge of the model skill to represent snow-forest interactions in regional or global applications.

The texture and thermal properties of the underlying soil can strongly impact the snowpack-ground heat exchange, snowpack
energy fluxes and snowpack dynamics (Decharme et al., 2016). Peatlands are rich in soil organic carbon (SOC) and charac-
terized by a high porosity, shallow water table, a weak hydraulic suction, strong gradient in hydraulic conductivity from high
values at the top to low values at the subsurface, low thermal conductivity, and large heat capacity (Decharme et al., 2016;
Marttila et al., 2021; Morris et al., 2022; Menberu et al., 2021). These properties result in a wet soil profile resistant to temper-
ature variations, while the drier top peat and moss layer can also provide effective insulation particularly during summertime
(Beringer et al., 2001; Park et al., 2018; Chadburn et al., 2015). The importance of the soil texture is still often overlooked even
in detailed snow models. For instance, in model comparisons of the ESM-SnowMIP project (Krinner et al., 2018), no SOC
information was used to parameterize the participating LSMs to the reference sites. In addition, many spatial snow simulations



neglect peat soils or SOC altogether, and their hydrological and thermal characteristics are derived from fractions of sand, silt and clay (Vernay et al., 2022; Brun et al., 2013; Mazzotti et al., 2021; Richter et al., 2021)

The goal of this study is to evaluate the ability of SURFEX LSM (Surface Externalisée, Masson et al., 2013) to describe the surface energy balance and its drivers in boreal and subarctic peatlands and forests. We evaluate the effect of turbulent exchange and snowpack parameterizations, and examine the skills of alternative model configurations to represent the soil-vegetation-snow interactions. The modelling framework includes flexible parameterizations for different processes within Crocus snowpack model (Vionnet et al., 2012), and its coupling to ISBA (Interactions between the Soil Biosphere and At-

mosphere, Noilhan and Mahfouf, 1996; Decharme et al., 2016) and MEB (Boone et al., 2017; Napoly et al., 2017) models enable assessments of soil-snow-vegetation interactions. We compare the model simulations against observed surface energy fluxes, snow depth and soil temperatures from two forest and two peatland sites in Finland. We focus on the snow cover period, but cover also the snow-free season for a reference. On the peatland sites, we test the sensitivity of the surface heat fluxes to different turbulence and snow parameterizations, and assess whether the SOC parameterization is necessary to reproduce soil

temperature and snowpack dynamics. On the forest sites, we compare the simulations of ISBA composite soil-vegetation and MEB big-leaf forest scheme to assess the suitability of different forest-snow model structures.

## 2   Materials and methods

### 2.1   Study sites

We consider coniferous forest and peatland ecosystems in southern and northern Finland. Both areas are located in the boreal biome and have seasonal snow cover (Fig. 1, Table 1).

#### 2.1.1   Pallas Supersite

The Pallas area represents northern subarctic conditions, and is characterized by pine and spruce forests, wetlands, fells and lakes (Aurela et al., 2015; Lohila et al., 2015; Marttila et al., 2021). In this study, we use eddy-covariance (EC) flux stations

on a pristine peatland (Lompolojänkkä, later denoted as northern wetland, N-WET), and on a mineral soil spruce forest (Kenttärova, northern forest, N-FOR). Both sites and their measurements have been described in detail by Aurela et al. (2015) and only the most salient features are reported here.

    Lompolojänkkä (67°59.835' N, 24°12.546' E) is a northern boreal mesotrophic sedge fen where the wetter parts are domi-

nated by sedges (*Carex rostrata* (most abundant), *Carex chordorrhiza*, *Carex magellanica* and *Carex lasiocarpa*) and the drier parts consist of shallow deciduous trees (*Betula nana* and *Salix lapponum*). Moreover, the fen has a fairly low coverage of





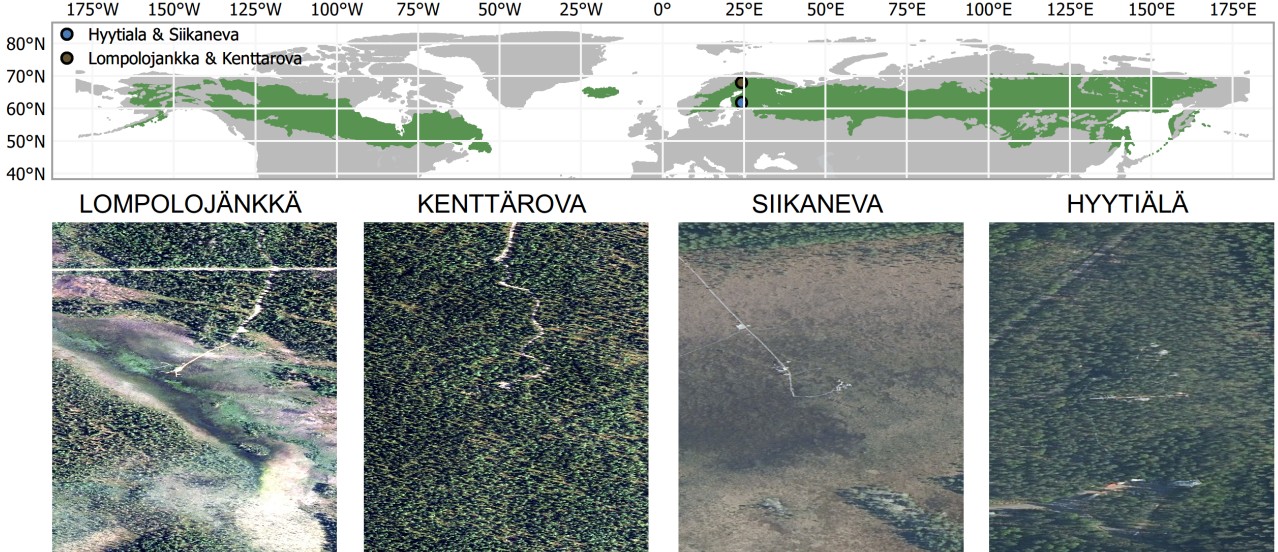

**Figure 1.** Study area locations inside the boreal land biome (green area, Olson et al., 2001) and aerial images of each site (NLSF, 2020).

shrubs, mainly *Andromeda polyfolio* and *Vaccinium oxycoccos*. The vegetation height is shallow (∼0.4 m), with exception of isolated trees/bushes on the drier edges of the peatland.

Kenttärova (67°59.237' N, 24°14.579' E) is a northern boreal spruce forest located on a hill-top plateau, approximately
60 meters above Lompolojänkkä wetland. The forest is dominated by Norway spruce (*Picea abies*) with some deciduous trees, mainly birch (*Betula pubescens*) but also aspen (*Populus tremula*) and pussy willow (*Salix caprea*). According to the classification by Brunet (2020), Kenttärova is a sparse forest.

### 2.1.2 Hyytiälä and Siikaneva

The sites are located in southern subarctic conditions in the Pirkanmaa region in southern Finland, at about 5 km distance
from each other. Siikaneva fen (southern wetland, S-WET), (61°49.961'N, 24°11.567'E) is a southern boreal oligotrophic fen dominated by sedges (*Eriophorum vaginatum*, *Carex rostrata* and *Carex limos*), and has an extensive Sphagnum cover (mainly *Sphagnum balticum*, *Sphagnum majus* and *Sphagnum papillosum*). The site has been described in detail in Aurela et al. (2007); Alekseychik et al. (2017).

Hyytiälä (southern forest, N-FOR; 61°50.471'N 24°17.439'E) is managed boreal Scots pine (*Pinus sylvestris*) dominated
forest on mineral soils, described in detail by Hari et al. (2013); Launiainen (2010); Launiainen et al. (2022). According to the classification by Brunet (2020), the site is a dense forest.



**Table 1.** General site information.

| Site | Code | Coordinates | Ecosystem | Soil type |
| --- | --- | --- | --- | --- |
| Lompolojänkkä | N-WET | 67°59.835' N, 24°12.546' E | mesotrophic fen | peat |
| Siikaneva | S-WET | 61°49.961'N, 24°11.567'E | oligotrophic fen | peat |
| Kenttärova | N-FOR | 67°59.237' N, 24°14.579' E | sparse spruce forest | podzol |
| Hyytiälä | S-FOR | 61°50.471' N 24°17.439' E | dense pine forest | podzol |

## 2.2 Models

We use components from the SURFEX (Surface Externalisée, Masson et al., 2013) modeling platform. SURFEX was selected as its modularity and vast range of model structures and incorporated process parameterizations enable its use in numerous applications to diverse research disciplines. Specifically, we used ISBA (Noilhan and Planton, 1989; Noilhan and Mahfouf, 1996) for composite soil-vegetation (on both peatland and forest sites), MEB (Boone et al., 2017; Napoly et al., 2017) for the canopy (on forest sites) and Crocus (Vionnet et al., 2012) and its ensemble/multiphysics version ESCROC (Lafaysse et al., 2017) for the snowpack simulations (all sites). In the next subsections, we briefly describe model components and parameterizations relevant to this study.

### 2.2.1 ISBA

ISBA (Interactions between the Soil Biosphere and Atmosphere) is the soil and vegetation component of SURFEX LSM (Noilhan and Planton, 1989; Noilhan and Mahfouf, 1996). It simulates the mass and energy fluxes in the soil-vegetation composite, as well as the exchanges between the soil-vegetation and the overlying atmosphere/snowpack (Fig. 2B,C). ISBA is used for the GCM by Meteo-France (Mahfouf et al., 1995; Douville et al., 1995; Salas-Mélia et al., 2005; Voldoire et al., 2013, 2019) and for NWP in numerous countries (e.g. Hamdi et al., 2014; Bengtsson et al., 2017).

In ISBA, the surface heat flux into the soil-vegetation composite (G, $\mathrm{Wm}^{-2}$) is computed as the residual of the sum of all surface/atmosphere energy fluxes:

$$G = R_n + H + LE \tag{1}$$

where $R_n$ ($\mathrm{Wm}^{-2}$) is the net radiation, H ($\mathrm{Wm}^{-2}$) is the sensible heat flux, and LE ($\mathrm{Wm}^{-2}$) is the latent heat flux. $R_n$ is the sum of the net shortwave radiation and the net longwave radiation:

$$R_n = R_g(1 - LSA) + \epsilon(R_A - \sigma T_s^4) \tag{2}$$

where $R_g$ ($\mathrm{Wm}^{-2}$) and $R_A$ ($\mathrm{Wm}^{-2}$) are the incoming shortwave and longwave radiations, respectively. The land surface albedo is denoted as LSA, and $\epsilon$ is the surface emissivity, $\sigma$ is the Stefan-Boltzman constant and $T_s$ (K) is the surface temperature. H



is computed with the bulk aerodynamics approach:

$$H = \rho_a c_\rho C_H V_a (T_s - T_a) \tag{3}$$

where the air density, the specific heat capacity, the wind speed and air temperature are denoted with $\rho_a$ (kgm$^{-3}$), $c_p$, (Jkg$^{-1}$K$^{-1}$), V$_a$ (ms$^{-1}$) and T$_a$ (K), respectively. C$_H$ is the turbulent exchange coefficient described later. When the soil is not covered by snow, LE is the sum of evaporation from the bare soil surface, E$_g$, evaporation of intercepted water on the

canopy, E$_c$, transpiration from the vegetation, E$_v$, and sublimation from bare soil ice, S$_i$:

$$LE = L_v (E_g + E_c + E_{tr}) + (L_f + L_v)(S_i) \tag{4}$$

where L$_v$ (Jkg$^{-1}$) and L$_f$ (Jkg$^{-1}$) are the latent heat of vaporization and fusion, respectively. Evaporation from the bare soil surface is computed as:

$$E_g = (1 - veg)\rho_a C_H V_a [h_u q_{sat}(T_s) - q_a] \tag{5}$$

where $veg$ is the fraction of vegetation cover, q$_{sat}(T_s)$ (kgkg$^{-1}$) is the saturated specific humidity at the surface, $q_a(T_s)$ (kgkg$^{-1}$) is the atmospheric specific humidity, h$_u$ is the dimensionless relative humidity at the ground surface related to the superficial soil moisture content. The sum of evaporation of canopy intercepted water (E$_c$) and transpiration (E$_{tr}$) is:

$$E_c + E_{tr} = veg\rho_a C_H V_a h_v [q_{sat}(T_s) - q_a], \tag{6}$$

where h$_v$ is the dimensionless Halstead coefficient describing the E$_c$ and E$_{tr}$ partitioning between the leaves covered and not

covered by intercepted water (see Noilhan and Mahfouf (1996) for details).

The turbulent exchange coefficient C$_H$ is based on the formulation of Louis (1979):

$$C_H = \left[ \frac{k^2}{ln(z_u/z_{0t}) ln(z_a/z_{0t})} \right] f(R_i) \tag{7}$$

where $z_u$ (m) is the reference height of the wind speed V$_a$ (ms$^{-1}$), $z_a$ (m) is the reference height of the air temperature and humidity, $z_{0t}$ (m) is the roughness height for heat, $k$ (-) is the von Karman constant and $f(R_i)$ (-) describes the decrease of C$_H$

as a function of increasing atmospheric stability, represented through Richardson number (R$_i$) (Louis, 1979).

Instead of separate treatment of the vegetation canopy and ground, ISBA considers the composite soil-vegetation energy budget (Fig. 2B,C). In the most detailed soil scheme ISBA-Diffusion (ISBA-DIF, Boone et al., 2000; Decharme et al., 2011), used in this study, 1D Fourier law is used to solve the soil heat diffusion, while a mixed-form Richards equation is applied for

the 1D soil water movements. Similar as in Napoly et al. (2020), we use the $A_{g-s}$ stomatal resistance formulation derived from the coupling of photosynthetic CO$_2$ demand and stomatal function (i.e. A-g$_2$s; Calvet et al. (1998)). ISBA uses parameters such as one-sided leaf area index (LAI, m$^2$m$^{-2}$), vegetation height, vegetation thermal inertia (Km$^2$J$^{-1}$), albedo of soil and vegetation, fractions of sand and clay as well as SOC content to characterize the composite soil-vegetation column. These



parameters may be defined by the user, or obtained from global or regional databases (e.g. Faroux et al., 2013) and pedotransfer

functions (Noilhan and Mahfouf, 1996; Peters-Lidard et al., 1998). In the presence of full snow cover, the surface energy budget is solved by Crocus (Sect. 2.2.3). For partial snow cover, Crocus is used to solve the snow covered fraction while the energy balance of the snow-free fraction is computed by ISBA, and total surface energy fluxes are computed as weighted averages of the snow and snow-free fractions (Sect. 2.3.1).

### 2.2.2 MEB

MEB (Multi-Energy Balance) is a recent ISBA development to explicitly describe vegetation and soil energy and mass balances. It was developed initially for forests (Boone et al., 2017; Napoly et al., 2017) and found to yield improved snow and soil temperature simulations (Napoly et al., 2020) but has not been evaluated for boreal and subarctic conditions. MEB simulates surface energy budget explicitly for the soil and vegetation canopy (a two-source model). When the ground is snow covered, the energy budget of the snowpack is also explicitly represented (i.e. a three-source model is applied). We used the MEB option,

where the forest floor is covered by a litter layer instead of the bare soil surface (Napoly et al., 2020) (Fig. 2A). MEB uses a big-leaf approach, meaning that the entire vegetation canopy is lumped into a single effective 'leaf' (Boone et al., 2017). The respective energy balance equations for the big-leaf canopy, the snowpack and the ground surface/litter layer in MEB are:

$$
\begin{cases}
C_v \frac{\partial T_v}{\partial t} = R_{nv} - H_v - LE_v + L_f \phi_v \\
C_{g,1} \frac{\partial T_{g,1}}{\partial t} = (1 - \rho_{sng})(R_{ng} - H_g - LE_g) + \rho_{sng}(G_{gn} + \tau_{n,Nn} SW_{nn}) - G_{g,1} + L_f \phi_{g,1} \\
C_{n,1} \frac{\partial T_{n,1}}{\partial t} = R_{nn} - H_n - LE_n - \tau_{n,1} SW_{nn} + \epsilon_{n,1} - G_{n,1} + L_f \phi_{n,1}
\end{cases}
\qquad (8)
$$

where $C_v$, $C_{g,1}$, $C_{n,1}$ ($\mathrm{Jm^{-2}K^{-1}}$) and $T_v$, $T_{g,1}$, $T_{n,1}$ K are the effective heat capacities and temperatures of the canopy,

ground surface/litter layer and snowpack, respectively. In these equations, the subscript 1 represents the uppermost layer or the base layer for the soil and the snowpack, respectively. $R_{nv}$, $R_{ng}$, $R_{nn}$ ($\mathrm{Wm^{-2}}$) are net radiation, i.e. the sum of net shortwave radiation and net longwave radiation from/to the corresponding layer. The shortwave radiation scheme used in MEB is described in detail in Carrer et al. (2013), and the H and LE flux parameterization as well as the stability correction function is detailed in Boone et al. (2017). Obviously $T_v$, $T_{g,1}$, $T_{n,1}$ are also involved in the radiative and turbulent terms, providing a

linear system of equations to be solved by an implicit numerical scheme. In this study, MEB is coupled to the ISBA-DIF soil scheme (Sect 2.2.1) and the snowpack model Crocus (Sect. 2.2.3). Energy fluxes between the canopy and the ground surfaces are calculated within MEB, and prescribed as upper boundary conditions in the subsequent Crocus and ISBA-DIF calculations.

### 2.2.3 Crocus

Crocus is a 1D physically based multilayer snowpack model. It is the most detailed snow scheme in ISBA, and has been used

for operational avalanche hazard forecasting in the French mountain ranges for the past three decades (Morin et al., 2020). It aims to mimic the vertical layering of snowpacks with a lagrangian discretization system, avoiding the aggregation of snow





layers with highly different physical properties. A detailed description of Crocus and its integration in SURFEX can be found in Vionnet et al. (2012).

In Crocus, the vertical heat diffusion in the snowpack is solved with an implicit backward-difference integration method (Boone
and Etchevers, 2001). The snow effective thermal conductivity, $k$, is based on Yen (1981):

$$k = k_{ice}(\frac{\rho}{\rho_w})^{1.88} \tag{9}$$

The snowpack surface net energy flux is the sum of net radiation, turbulent fluxes and advective fluxes from precipitation. Over the snow, the sensible heat flux is computed similarly as in ISBA (Eq. 3) for soil surface, while the latent heat flux (sublimation/deposition), $LE_s$, is computed as:

$$LE_s = (L_f + L_v)\rho_a C_H U[q_{sat}(T_s - q_a)], \tag{10}$$

where $T_s$ $(K)$ is the snow surface temperature. The bottom of the snowpack and the uppermost soil layer are fully coupled with a mass and energy-conserving semi-implicit solution. The heat conduction $G$ at the snow-soil interface is explicitly computed, and depends on the temperature gradient between the bottom snow layer and the uppermost soil layer.

## 2.3 Model configurations and parametrization

### 2.3.1 Model configurations

We use three different configurations of ISBA, MEB and Crocus modules (Fig. 2). The first configuration (Fig. 2A) is the big-leaf approach where the fluxes between the canopy and ground are explicitly computed by MEB, and prescribed in the subsequent Crocus snowpack and ISBA-DIF soil modules (later denoted as MEB).

The two other configurations use the composite soil-vegetation conceptualization of ISBA (Fig. 2B,C), and differ only in how the snow cover fraction is represented over the soil-vegetation composite. ISBA aggregates the properties of soil and vegetation depending on the vegetation fraction (veg) that covers a given grid-cell. Then, a dynamic snow fraction determines the part of the soil-vegetation composite that is covered by snow while the remaining (non snow) soil-vegetation fraction stays in constant contact with the atmosphere. This model version is later denoted as ISBA varying snow cover (ISBA-VS, Fig. 2B).
The effective snow cover fraction is defined as the average between the snow fraction of vegetation ($p_{snv}$) and snow fraction of the ground ($p_{sng}$), calculated as (Decharme et al., 2019; Napoly et al., 2020):

$$p_{sn} = veg\,p_{snv} + (1 - veg)\,p_{sng} \tag{11}$$

$$p_{snv} = min(1.0, \frac{HS}{HS + w_{sw}z0}) \tag{12}$$



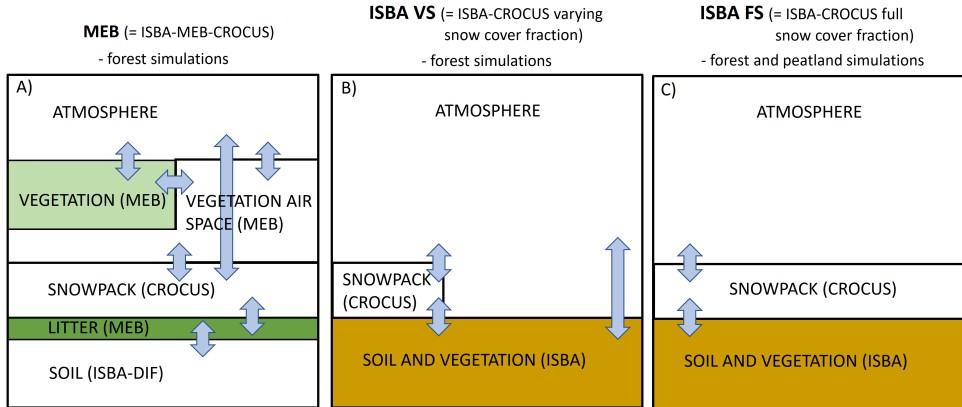

**Figure 2.** Three different model configurations used in the study: (A) MEB big-leaf approach (= ISBA-MEB-Crocus), (B) ISBA-VS approach with varying snow cover fraction over the soil-vegetation composite, and (C) ISBA-FS approach with full snow cover fraction over the soil-vegetation composite. Energy flux exchanges are represented with arrows.


$$p_{sng} = min(1.0, \frac{HS}{HS_g})  \qquad (13)$$

where HS $(m)$ is the height of the snow, $HS_g$ is the threshold value for height of the snow (0.01 m by default), and z0 $(m)$ denotes the surface roughness. The coefficient $w_{sw}$ relates to vegetation characteristics and is assigned as 5 by default in SURFEX and in NWP configurations (as well as in this study). However, without clear consistency, different values of $w_{sw}$

have been used e.g. in climate simulations ($w_{sw}$ = 2 by Decharme et al., 2019) and hydrological applications ($w_{sw}$ = 0.2 by Le Moigne et al., 2020). As documentation about the snow cover fraction and the parameter $w_{sw}$ is widely lacking, we present a summary of the application specific treatment of the snow cover fraction and $w_{sw}$ in Appendix D (Table D1).

The third configuration is the common approach for open site snow simulations (Vernay et al., 2022; Nousu et al., 2019)
and for some large scale reanalyses (Brun et al., 2013). It assumes that the snowpack is fully covering the soil-vegetation composite, and snow cover fraction is unity regardless of the snow depth (Fig. 2C). This version is later denoted as ISBA full snow cover (ISBA-FS). It has never been used in coupled applications with atmospheric models but frequently in hydrological applications (Lafaysse et al., 2011; Revuelto et al., 2018). Most site-level evaluations of SURFEX snow schemes also rely on ISBA-FS configuration (Decharme et al., 2016; Lafaysse et al., 2017).

**2.3.2 ESCROC parameterizations for snow processes and turbulent exchange**

We use the multiphysics version of Crocus (ESCROC, Ensemble System Crocus, Lafaysse et al., 2017) to evaluate the impact and associated uncertainties of the different parameterizations of snow processes and turbulent exchange. In ESCROC, the





main physical processes and properties of snowpack, as well as the turbulent fluxes, can be represented by several alternative options. Different options are available to parameterize density of new snow, snow metamorphism, absorption of solar radiation, turbulent fluxes, thermal conductivity, liquid water holding capacity, snow compaction and surface heat capacity. Lafaysse et al. (2017) have shown that consideration of all these combinations is numerically expensive and often unnecessary to depict the overall uncertainty. Indeed, an optimized standard subensemble of 35 members ($E_2$ subensemble) has been found sufficient to provide a spread of the appropriate magnitude compared to model errors (Lafaysse et al., 2017). In this work we used the $E_2$ subensemble, similar to recent studies quantifying the model uncertainty (e.g. Deschamps-Berger et al., 2022; Tuzet et al., 2020). In our case, the presented ensemble spread correspond to simulated values between ensemble minimum and maximum.

In Crocus, the default turbulent exchange parameterization (Eq. 7) is expected to underestimate the turbulent fluxes under stable conditions (Martin and Lejeune, 1998). Therefore, different stability dependencies of the $C_H$ have been implemented in ESCROC. They differ mainly in the $R_i$ thresholds below which $C_H$ is assigned a constant value to enable turbulent heat and mass transport under stable conditions. As shown by Fig. 4 in Lafaysse et al. (2017), these parameterizations are a) classical Louis (1979) formula (later referred as RIL) with threshold at $R_i = 0.2$, b) RIL with threshold at $R_i = 0.1$ (RI1), c) RIL with threshold at $R_i = 0.026$ (RI2), and d) modified formulation with effective roughness length for heat ($10^{-3}$m), minimum wind speed (0.3 ms$^{-1}$), and with threshold at $R_i = 0.026$ (M98) by Martin and Lejeune (1998). Although the RIL parameterization is widely used in SURFEX applications (e.g. Decharme et al., 2019; Le Moigne et al., 2020), the RI2 parameterization of is applied in operational snow modelling in the Alpine area (Vernay et al., 2022), and M98 was recently used in the Canadian Arctic by Lackner et al. (2021). However, evaluations of the different Crocus turbulent flux parameterizations against surface flux data are lacking. MEB uses a different stability correction term (Boone et al., 2017) and applies only the RIL option for the stable conditions. While the MEB simulations (with ESCROC) are based on the $E_2$ subensemble as well, they therefore only use the RIL turbulent exchange parametrization for all members.

### 2.3.3 Site parameters

The parameterization of ISBA and MEB for the study sites is given in Table 2. Summer LAI and vegetation height were obtained from literature, while winter LAI was estimated according to the proportion of deciduous and coniferous vegetation on each site. The LAI of S-FOR refers to conditions before forest thinning in early 2020. The thinning, resulting in ca. 35% reduction in LAI, was neglected in our simulations as major part of the simulation period covers time before the thinning. Vegetation types in ISBA are characterized according to ECOCLIMAP (Champeaux et al., 2005); the forest sites in this study classify as boreal needleleaf evergreen (BONE), while the peatland sites are best represented as boreal grass (BOGR). Additional parameters based on LAI, vegetation height and vegetation type are computed following the standard methods of ISBA (Noilhan and Mahfouf, 1996; Carrer et al., 2013).

Soil texture (sand and clay fractions) for the forest sites are based on *in situ* measurements. The peatland SOC have not been measured, but we assumed the top 1 m of the peatlands to consist 100 % of SOC, while the deeper layers were assigned



**Table 2.** Main model parameters for the study sites. Vegetation types BOGR and BONE correspond to boreal grass and boreal needleleaf evergreen, respectively.

| Parameter | N-WET | S-WET | N-FOR | S-FOR | Source |
|---|---|---|---|---|---|
| Veg. type | BOGR | BOGR | BONE | BONE | ECOCLIMAP: Champeaux et al. (2005) |
| Veg. fraction (only with ISBA) (-) | 0.95 | 0.95 | 0.95 | 0.95 | ECOCLIMAP: Champeaux et al. (2005) |
| Veg. height (m) | 0.4 | 0.25 | 13 | 15 | Aurela et al. (2015); Alekseychik et al. (2017) Kolari et al. (2022) |
| $LAI_{max}$ ($m^2 m^{-2}$) | 1.3 | 0.6 | 2.1 | 3.0 | Aurela et al. (2015); Alekseychik et al. (2017) Kolari et al. (2022) |
| $LAI_{min}$ ($m^2 m^{-2}$) | 0.3 | 0.1 | 1.9 | 2.4 | assigned |
| Veg. albedo (NIR/VIS) (-) | 0.136 | 0.187 | 0.145 | 0.145 | assigned |
| Soil albedo (NIR/VIS) (-) | 0.136 | 0.187 | 0.145 | 0.145 | assigned |
| Tair measurement height (m) | 2 | 2 | 2 | 2 | FMI (2021) |
| Wind measurement height (m) | 13 | 16.8 | 23 | 16.8 | Aurela et al. (2015); Mammarella et al. (2019) Alekseychik et al. (2022a) |
| Elevation (m) | 270 | 162 | 347 | 181 | Hari et al. (2013); Alekseychik et al. (2022a); FMI (2021) |
| Clay (%) | 9 | 7 | 9 | 7 | measurements |
| Sand (%) | 76 | 65 | 76 | 65 | measurements |
| SOC (0-30cm) (% / $kgm^{-2}$) | 100 % | 100 % | 3.0 $kgm^{-2}$ | 3.5 $kgm-2$ | assigned / Lindroos et al. (2022) |
| SOC (30-70cm) (% / $kgm^{-2}$) | 100 % | 100 % | 1.75 $kgm^{-2}$ | 0.75 $kgm^{-2}$ | assigned / Lindroos et al. (2022) |
| SOC (70-100cm) (% / $kgm^{-2}$) | 100 % | 100 % | 0 $kgm^{-2}$ | 0 $kgm^{-2}$ | assigned |
| Start of simulation (yyyy-mm) | 2013–09 | 2016–09 | 2013–09 | 2008–09 | - |
| End of simulation (yyyy-mm) | 2021–07 | 2021–07 | 2021–07 | 2021–07 | - |

a mineral soil similar to the contiguous forests. Although peat profiles may be deeper, the soils below the damping depth of annual temperature fluctuations (ca 1.1 m for saturated peat soil) are assumed not to have significant impact on surface energy flux dynamics. SOC values for mineral soils of N-FOR and S-FOR were taken from Lindroos et al. (2022). The rest of the

295 parameters presented in Table 2 were assigned as estimates. The detailed thermal and water retention parameters are derived based on the soil texture using the pedotransfer functions of ISBA (Noilhan and Mahfouf, 1996; Peters-Lidard et al., 1998).

## 2.4 Data

### 2.4.1 Model forcing

Meteorological forcing consist of hourly observations of $T_a$, $V_a$, precipitation rate ($P$), $q_a(T_s)$, $R_g$ and $R_A$ and atmospheric pressure. The available meteorological observations from the nearest meteorological stations were obtained from Finnish Me-



teorological Institute (FMI) open database (FMI, 2021) (Station IDs: N-WET 778135, N-FOR 101317, S-FOR 101987). Meteorological observations at the S-WET site come from the SMEAR database (Alekseychik et al., 2022a; Mammarella et al., 2019). At S-WET and S-FOR the shortwave and longwave radiation were obtained from the SMEAR database, while at N-WET and N-FOR data from FMI stations were used. The diffuse to total shortwave radiation ratio, $r$, was estimated as a function of the cosine of the sun zenith angle, $\mu$. More specifically, a 3rd degree polynomial fit between $r$ and $\mu$ was obtained using the atmospheric model SBDART (Ricchiazzi et al., 1998) to simulate diffuse and total solar radiation in clear sky conditions. The atmospheric profile was set to typical winter conditions, 0.09 for the aerosol optical thickness, 300 DU for the ozone column and 0.854 $\mathrm{gcm}^{-2}$ for the water vapor column.

The data gaps in meteorological observations were first filled by the contiguous sites and the remaining gaps by other nearby meteorological stations (IDs: N-WET/N-FOR 101932, S-WET/S-FOR 101520). The missing radiation observations were first filled by the contiguous sites, and the remaining gaps by ERA5 reanalysis data (Hersbach et al., 2020). Furthermore, the observations of $P$ were split into snow and rain based on $T_a$:

$$
P : \begin{cases} P_{ice} & \text{if } T_a \leq 0°\text{C} \\ P_{liq} & \text{if } T_a \geq 1°\text{C} \\ aP_{ice} + bP_{liq} & \text{for } 0°\text{C} < T_a < 1 \text{ °C} \end{cases} \tag{14}
$$

where $P_{ice}$ and $P_{liq}$ denote the snowfall and rainfall rates, respectively. Between 0°C and 1°C the fraction of ice/snow changes linearly ($a$ = 1 - $b$).

### 2.4.2 Model evaluation data

We use surface energy flux observations, height of snow (HS) and soil temperatures in model evaluation. The availability period of each variable is given in Appendix A (Table A1). On all sites, reflected shortwave radiation (SWU) and outgoing longwave radiation (LWU) were measured using pyranometers and pyrgeometers, while ground heat flux (G) was measured using soil heat flux plates between 5 and 10 cm depths.

The H and LE were measured by the eddy-covariance (EC) technique. The EC systems consist of USA-1 (METEK) three-axis and Gill HS-50 sonic anemometers as well as closed-path LI-7000 and LI-7200 (Li-cor, Inc.) $CO_2$/$H_2O$ analysers (Aurela et al., 2015; Mammarella et al., 2016; Alekseychik et al., 2022b). The detailed procedure for obtaining the turbulent heat fluxes from raw eddy covariance data is detailed in Aurela et al. (2015); Mammarella et al. (2016). In short, the sensible and latent heat fluxes were screened for instrument failure and data outliers, and data quality flags were made according to friction velocity ($u_*$) and flux stationarity (FST) criteria (Foken et al., 2005):

- flag 2: all data (after screening of instrument failures and outliers)

- flag 1: $u_* \geq 0.1 \text{ ms}^{-1}$ and $0.3 \leq \text{FST} \leq 1.0$

- flag 0: $u_* \geq 0.1 \text{ ms}^{-1}$ and $\text{FST} \leq 0.3$



At S-WET, S-FOR and N-FOR, Automated HS observations are directly used (FMI, 2021). On N-WET the automated HS measurement is at 0.7 m height, and therefore the exceeding snow depths were taken from biweekly manual measurements. To account for the spatial variability of snow depth in the forests, manual HS measurements from a snow course in the close proximity of the automated measurements were used (Aalto et al., 2022; Marttila et al., 2021). Each site has different configuration

of soil temperature sensors. At N-FOR and N-WET stations, soil temperatures are measured at 5, and 20 cm depths (Aurela et al., 2015). Soil temperatures at S-FOR and S-WET are measured at depths of 0, 5, 10, 30, 50 and 75 cm (Aalto et al., 2022).

## 2.5    Model experiments

On the peatland sites, we evaluate the skill of ISBA-FS (Sect. 2.3.1) and effect of ESCROC parameterizations (Sect. 2.3.2) on surface heat fluxes over snowpack and bare ground. The simulations are further used to assess the differences in HS and

soil temperature between ESCROC turbulent exchange options. For a more detailed evaluation of two contrasting turbulent exchange options within ESCROC, we conducted deterministic (ISBA-FS) simulations with i) all default ESCROC parameterization as in Fig 2. in Lafaysse et al. (2017) (processes listed in Sect. 2.3.2, referred as RIL), and ii) all the default ESCROC parameterizations except the turbulent exchange option switched to M98 (referred as M98). Moreover, we explore the influence of soil texture on the soil thermal regime and on snowpack dynamics. Hence, an additional deterministic simulation was

conducted where the soil was characterized as mineral while turbulent exchange was set to M98 (referred as MINERAL). The MINERAL simulation was compared to the previously described M98 simulation, where the soil was characterized as 1 m deep 100 % SOC (Table 2).

On the forest sites, we examine the skills of the different alternatives to represent the energy and mass budgets of soil and

vegetation (ISBA-VS, ISBA-FS, MEB in Sect. 2.3.1), and their implications on HS, soil temperature and surface energy fluxes. First, we conduct ESCROC simulations with these three configurations to focus on the HS and soil temperature. The ISBA-VS simulations are conducted with the default snow cover fraction parameterization (Eq. 11). For a more detailed comparison of the simulated and observed above-canopy surface energy fluxes by ISBA-VS and MEB, we conducted deterministic simulations with the default Crocus parameterizations (as in Fig 2. in Lafaysse et al. (2017)).


Model simulation periods for each site are in Table 2. For each site, the model initial state was obtained by a spin-up simulation from the start date (Table 2) to September 2020. In total of ca. 290 simulations were conducted, including all the ensemble and deterministic simulations.

## 2.6    Model evaluation metrics

Time series plots of daily averaged variables are used to represent the results, whereas mean absolute error (MAE), mean bias error (MBE) and coefficient of determination ($R^2$) are used in quantitative model-data comparison. To detect possible biases in model simulations, we use scatter plots and quantile-quantile plots of sorted observations against sorted simulations. The sign convention is so that the surface energy fluxes are presented relative to the surface (i.e. negative flux means that surface





is losing energy). In time series plots, the turbulent flux observations include all EC data (Sect 2.4.2, quality flag $\leq$ 2). For

the scatter and quantile-quantile plots, only flux data with quality flag $\leq$ 1 is used, and the results computed as aggregated

6-hour means include the full periods where simulations and observation are available (referred later as evaluation period). We

compare snow and snow-free conditions by grouping the results into time windows where models and observations agree of

the ground conditions (snow or snow-free).

## 3   Results

**3.1   Observed energy balance at peatland and forest sites**

The energy budget at high-latitudes have a strong seasonal variation driven by solar radiation (Fig. 3). In winter (December,

January, February), longwave radiation balance to large extent determines $R_n$, particularly in the northern Finland. As outgo-

ing longwave radiation (LWU) usually exceeds incoming, daily average $R_n$ is negative down to -50 $\mathrm{Wm}^{-2}$ and lower, which

implies considerable radiative cooling. Towards spring the radiation budget is gradually counterbalanced by shortwave radi-

ation. On the peatlands, a large fraction of SWD is reflected during snow cover, and daily $R_n$ turns positive in late melting

season (Fig. 3A,C). At the forest sites, the timing of $R_n$ becoming positive is less sensitive to the presence of snow on the

ground as a large proportion of the SWD is absorbed by the vegetation. In summer, high solar elevation and the absence of the

reflective snow surface cause daily $R_n$ to be up to 200 $\mathrm{Wm}^{-2}$, warming the ground and vegetation. After the summer solstice,

the shortwave radiation decreases, longwave radiation becomes gradually more dominant for the radiation balance, and the $R_n$

falls towards negative values in the autumn.

    The $R_n$ is balanced by H and LE, and snowpack/ground heat flux (Fig. 3). The residual line represents the amount of energy

that would be required to close the observed energy budget (Fig. 3). It includes changes in internal energy of the snowpack and

vegetation, but also reflects the common energy balance closure problem in EC-measurements (Mauder et al., 2020) (see Sect.

4.4). The energy balance closure in snow-free conditions was typical for EC-measurements, ranging from 0.81 to 0.99. The

lack of snowpack heat flux and/or temperature profile measurements did not enable assessing the closure during snow cover

periods. In winter, LE and G are small and the radiative cooling is counterbalanced mostly by H, corresponding to warming of

the snowpack and/or vegetation, and cooling of the ambient air. The mean Bowen ratios ($\beta$ = H/LE) during snow cover season

are high (N-WET = 5.9, S-WET = 6.3, N-FOR = 7.4, S-FOR = 2.7). The $R_n$ during winter falls lower (more negative) on the

northern sites, and thus also downward H becomes stronger (daily average up to 50 $\mathrm{Wm}^{-2}$). As $R_n$ increases in spring, the en-

ergy balance residual term increases as well. In summer, both H and LE are negative (upwards) heating the atmosphere, while

downward G drives the warming of soil profile. At all sites, LE increases along the growing season and peaks approximately

in July. In autumn, the turbulent fluxes decrease as response to reduced $R_n$. The mean $\beta$ over the snow-free season are N-WET

= 0.5, S-WET = 0.6, N-FOR = 1.2 and S-FOR = 0.9.






**Figure 3.** Daily averaged radiation budget (left) and surface energy budget (right) of hydrological year of 2018–2019. Colored stacks represent the observed fluxes relative to the surface as shown in legends (i.e. incoming fluxes are positive and outgoing fluxes negative). Dashed line in energy budget plot corresponds to the residual after the sum of each energy component whereas the dashed line in the radiation plot shows the net radiation ($R_n$). Note different scale in left and right columns. Ground heat flux (G) is missing on N-WET. The observed evolution of the height of snow (HS) is shown in gray polygon (not in scale).





**Figure 4.** Time series of daily averaged surface heat flux spread simulated by ESCROC 35 ensemble members against corresponding observed values during 2018–2019 snow season. H, LE and LWU correspond to sensible heat, latent heat and upward longwave radiation fluxes, respectively. The observed and simulated evolution of height of snow (HS) are shown in gray.

## 3.2 Peatland simulations

The sensitivity of surface heat fluxes and height of snow to different ESCROC model parameterizations is shown in Fig. 4. The spread corresponds to the difference between the minimum and maximum of the ensemble. Notably, H has relatively high spread and the observed H often lies near the limit or even outside the simulated range at both S-WET and N-WET. Modelled wintertime LE is low and, as for H, the observed values are near the limit or outside the simulated range, especially in spring. LWU has strong day to day variation well captured by the model, and the spread is rather small relative to the total flux.



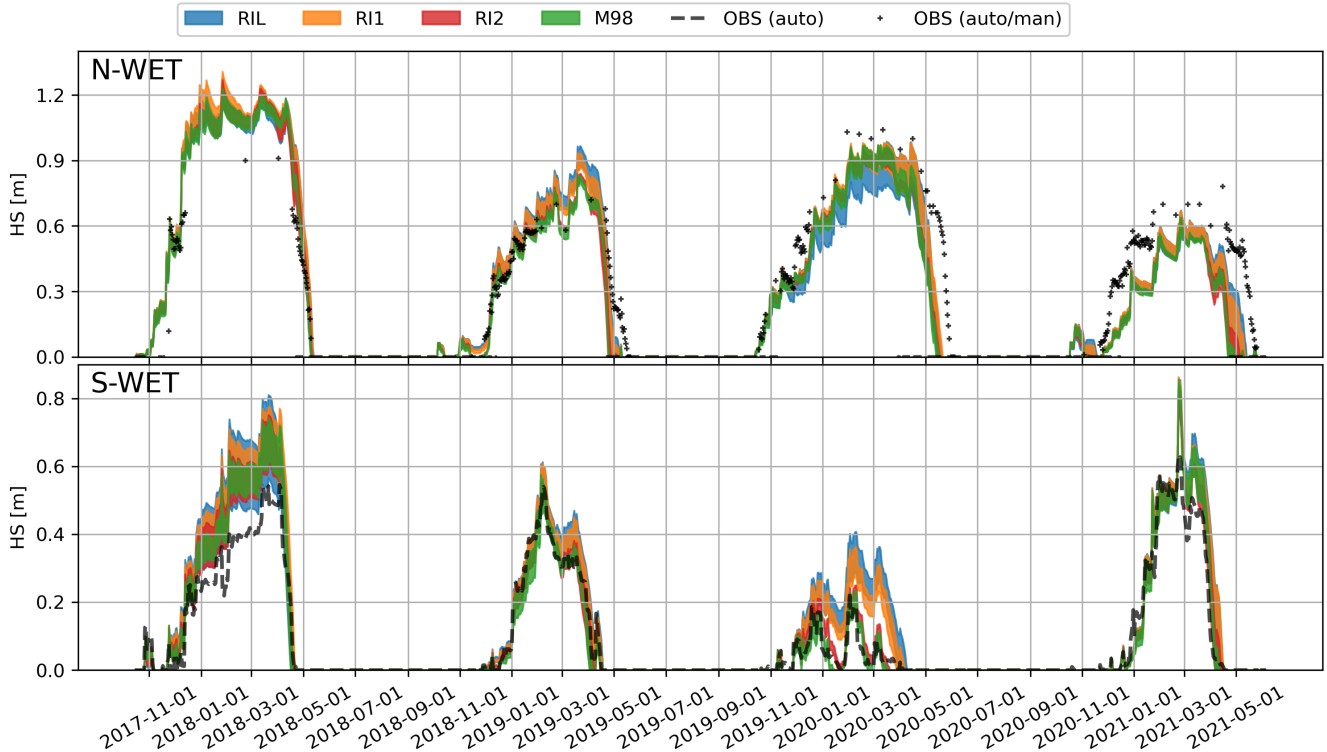

**Figure 5.** Time series of snow depths simulated by ISBA-ESCROC. The 35 ensemble members are grouped by their turbulent flux parameterization, and the spread of each group is presented in colored ranges. Observed snow depths are presented in black dots and dashed lines.

### 3.2.1 Impact of alternative turbulence ($C_H$) parameterizations

To assess the sensitivity of HS simulations to alternative turbulence parameterizations, and to alternative snow process options,

we examined the snow depth simulations where the ESCROC members are grouped according to their turbulent flux option (Fig. 5). The groups are consistently overlapping during snow accumulation periods on both sites, indicating that the differences in snow accumulation and maximum snow depth are driven mostly by the uncertainty of snow process descriptions. The groups diverge during snow melt, especially during winter melt events (e.g. 2019–2020 on S-WET), indicating higher importance of turbulent fluxes on snow dynamics. While it is difficult to identify a group that fits observed snow depths on the

N-WET site best, the winter melt event in 2018–2019 is only captured by the M98 and RI2 parameterizations. Likewise, at the S-WET site only RI2 and M98 are able to simulate the observed melt events.

This finding is consistent with comparison of simulated surface heat fluxes by the two deterministic runs (RIL and M98 in Sect. 2.5) against observations (Fig. 6). With the RIL parameterization, the magnitude of H and LE is largely underestimated,

while this bias is to most extent corrected by using M98. Improved simulation of H and surface temperature also entail im-





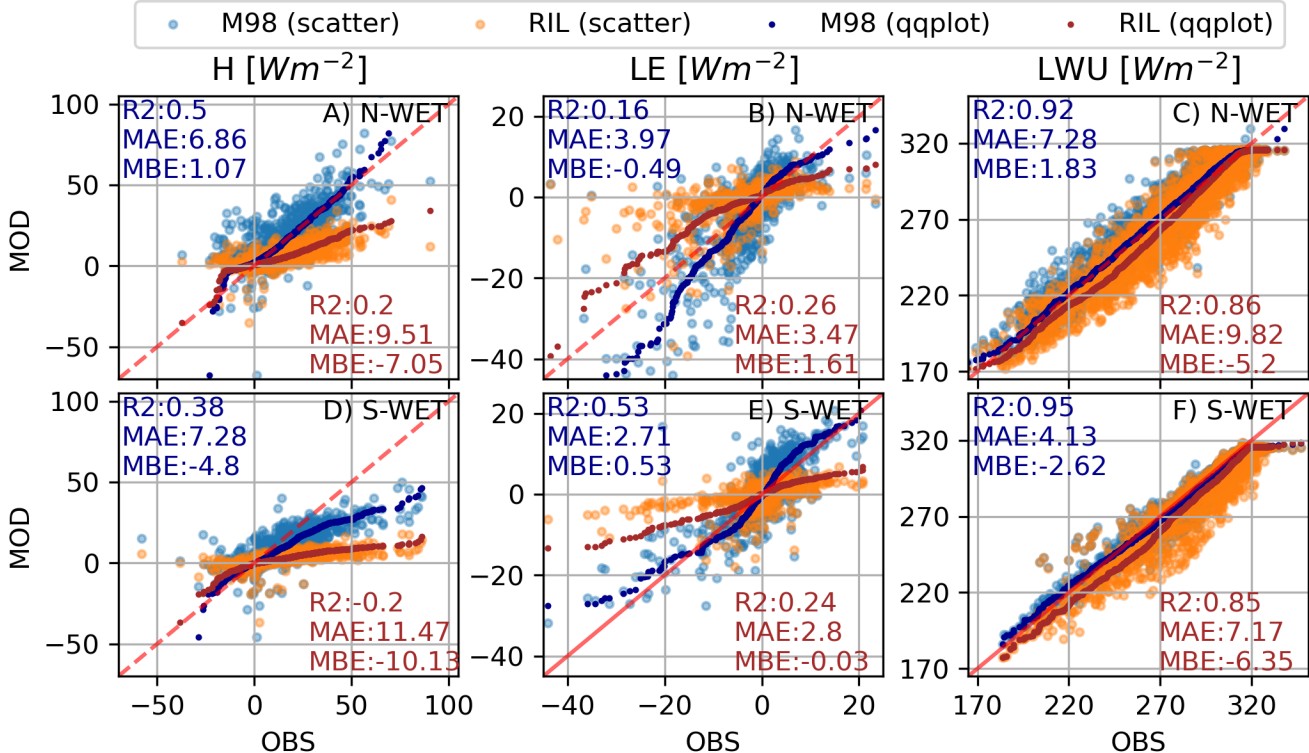

**Figure 6.** Scatterplots and quantile-quantile plots of surface heat fluxes during snow cover for the evaluation period with the RIL and M98 turbulence parameterizations.

proved LWU (Fig. 7). However, regardless of the major improvement, the modeled fluxes still only moderately correlate ($R^2$) with observations.

### 3.2.2 Radiative fluxes

We compare the simulated and observed LSA, SWU, LWU and surface temperatures with snow-free and snow conditions in Fig. 7. These experiments correspond to the deterministic M98 simulation as in Sect. 2.5.

The higher LSA during snow cover leads to much higher SWU during winter and spring than during summer. The modelled SWU generally well matches the observations, but the scatter increases with increasing SWD, indicating uncertainties in simulated LSA when shortwave forcing is high over the snowpack in spring. These cause a slight underestimation of simulated 425 spring LSA, also visible in the time series especially on N-FOR (Fig. 7). Moreover, simulated LSA tends to be overestimated during shallow snow depth both in spring and autumn (Fig. 12). This is because the ISBA-FS approach assumes snow to completely cover the ground regardless of the snow depth, while in reality the fractional snow cover can lower the LSA. The mean







**Figure 7.** Scatter plots and quantile-quantile plots of modelled against observed upward shortwave radiation (SWU) (A,B) and upwards longwave radiation (LWU) (C,D) on peatland sites with snow cover (w/ snow) and without snow cover (w/o snow) as well as the time evolution of 5-day rolling means of LSA and surface temperature (Ts) as simulated and observed from September 2018 to September 2019 (E,F). The evolution of the height of snow (HS) is not in scale.

absolute errors in simulating SWU are small and of similar magnitude (from ∼4 to 7 Wm⁻²) both for snow and snow-free conditions.


Warmer surface temperatures during snow-free season result in higher LWU compared to winter and spring (Fig. 7). The surface temperatures and LWU are generally well simulated across sites and ground conditions at least with the presented time intervals. During snow cover, the upper tail of the radiation distribution is slightly higher than simulated; however the mean biases are generally very low. There are no other visible biases in LWU simulations and the other metrics are also very good,





consistent with Fig. 4. The mean absolute errors in simulating LWU are similar for snow and bare ground, about (∼4 to 7 Wm$^{-2}$).

### 3.2.3   Soil thermal regime

The effect of soil parameterization on simulated soil temperature and HS dynamics at S-WET is shown in Fig. 8. Due to shallow water table, the soil profile remains nearly saturated throughout the year. As the porosity and field capacity in the SOC

parameterization are much higher than in the MINERAL, the former has also significantly higher heat capacity and smaller thermal diffusivity. This means soil temperature variations became more rapidly attenuated, and the effect becomes increasingly important in deeper soil layers (Fig. 8). The results show that including a realistic soil profile (SOC) greatly improves the peatland soil temperature simulations at depths 50–70 cm, but only slightly close to the surface (0–10 cm). On both sites, the simulated surface soil temperature variations in summer are greater than observed. This is presumably because ISBA does not

include the insulating moss/litter layer on top of the peat soil. Due to the weak influence on the surface soil temperatures, the soil parameterization (SOC vs. MINERAL) does not significantly affect the simulated snow depth (Fig. 8A), except during the low snow depths in winter 2019–2020. Simulated snow depth at N-WET was not sensitive to soil parameterization (not shown), likely because of the thicker snowpack and lack of melting-freezing cycles during the winter.

## 3.3   Forest simulations

### 3.3.1   Impact of vegetation representation on snow depth

The three different vegetation representations (Sect. 2.3.1) have highly contrasted effect on the forest energy budget, snowpack, and soil temperature simulations. In general, the snowpack simulations for the forest sites are poorer than for the peatland sites; however the observed snow depths also vary considerably within the forests (see OBS in Fig. 9 and Sect. 4.4).

The simulated snow depth with with the ISBA-VS (composite soil-vegetation and varying snow cover fraction, Fig. 2B) does not agree with the observations; the model version heavily overestimates accumulation and predicts extremely rapid, strong and too early melt events both at S-FOR and N-FOR (Fig. 9). Replacing the default snow cover fraction parameter (w$_{sw}$ = 5) with w$_{sw}$ = 0.2 (used for hydrological modelling in Le Moigne et al., 2020) yields slightly better HS dynamics for N-FOR, but the results remain unsatisfactory. The effect of snow cover fraction parameter w$_{sw}$ (see Sect. 2.3.1) for ISBA-VS snow depth

simulations is detailed in Appendix D (Fig. D1).

The MEB (explicit canopy, ground and snowpack energy balance) simulates the snow accumulation periods at N-FOR very well but peak snow is reached too early and maximum snow depths underestimated, due to combined impact of overestimated compaction and too early start and progression of the snow melt. The role of both processes was evident also from comparison of modeled and observed snow water equivalent (not shown).

ISBA-FS performs better during the snow accumulation period, with simulated snow depths very close to observations. However, the ablation of snow is too rapid, and the final melt out dates are close to those simulated by MEB. On the S-FOR



**Figure 8.** The effect of soil parameterization on simulated and observed height of snow (HS, A), and soil temperature profiles (B-G) during one hydrological year at the S-WET site. MINERAL refers to mineral soil and SOC to peat soil. The soil depths of measurements and simulations are presented in each panel.



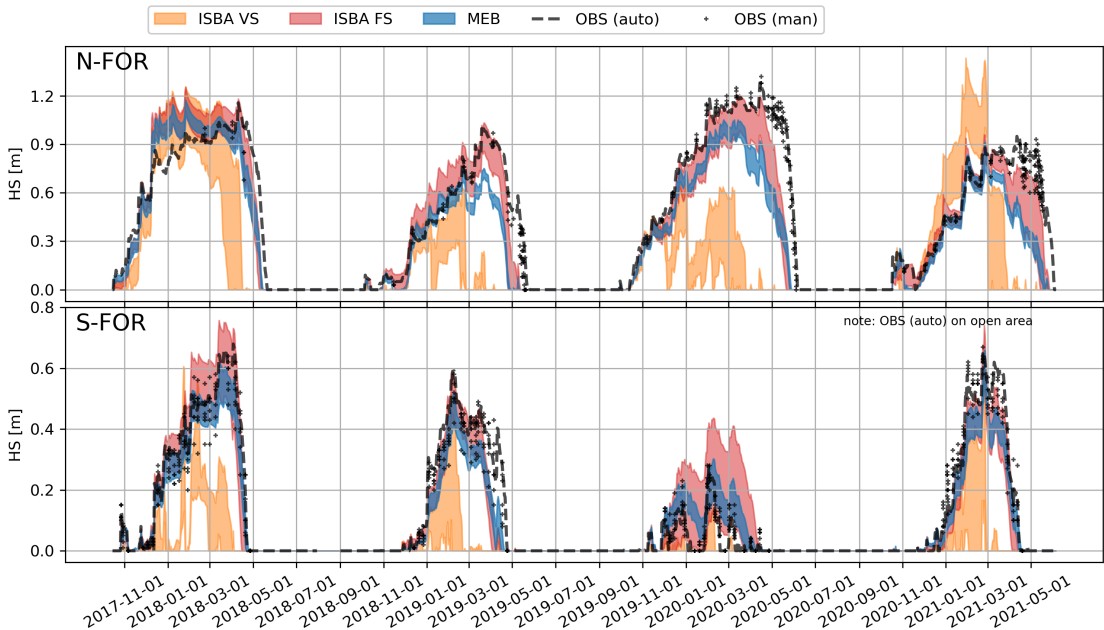

**Figure 9.** Effect of alternative configurations of ISBA and MEB on height of snow (HS). The envelopes visualize the corresponding ensemble spreads between minimum and maximum values.

site, MEB captures both snow accumulation (including peak snow depths), melt dynamics and final melt out dates rather well. ISBA-FS predictions are generally close to MEB. As MEB only considers one option for turbulent exchange (RIL), the spread of the ensemble is smaller than for the ISBA configurations (Fig. 9). The uncertainties of other snow processes accounted

for in ESCROC are not sufficient to explain the discrepancies between simulated and observed snow depths, suggesting that uncertainties in the canopy process representations prevail in these simulations.

### 3.3.2   Impact of vegetation representation on soil temperature

Similar to the snow depth, soil temperature predictions by ISBA-VS are erroneous, with drastically underestimated temperatures and unrealistic dynamics at all depths. While MEB and ISBA-FS provided very similar snow depth, the soil temperatures

simulated by MEB agree better with the observations although there is a cold bias in autumn and a warm bias in summer (Fig. 10). On N-FOR the warm bias in winter by MEB may be important for determining the soil frost regime. Interestingly, ISBA-FS seems to capture the winter soil temperatures better on N-FOR, but this may be due to the larger cold bias in autumn. All model versions tend to overestimate day-to-day temperature variability.



**Figure 10.** Effect of alternative configurations of ISBA and MEB on soil temperature profile. The envelopes visualize the corresponding ensemble spreads. The observed evolution of the height of snow (HS) is not in scale. The soil depths of measurements and simulations are presented in each panel.





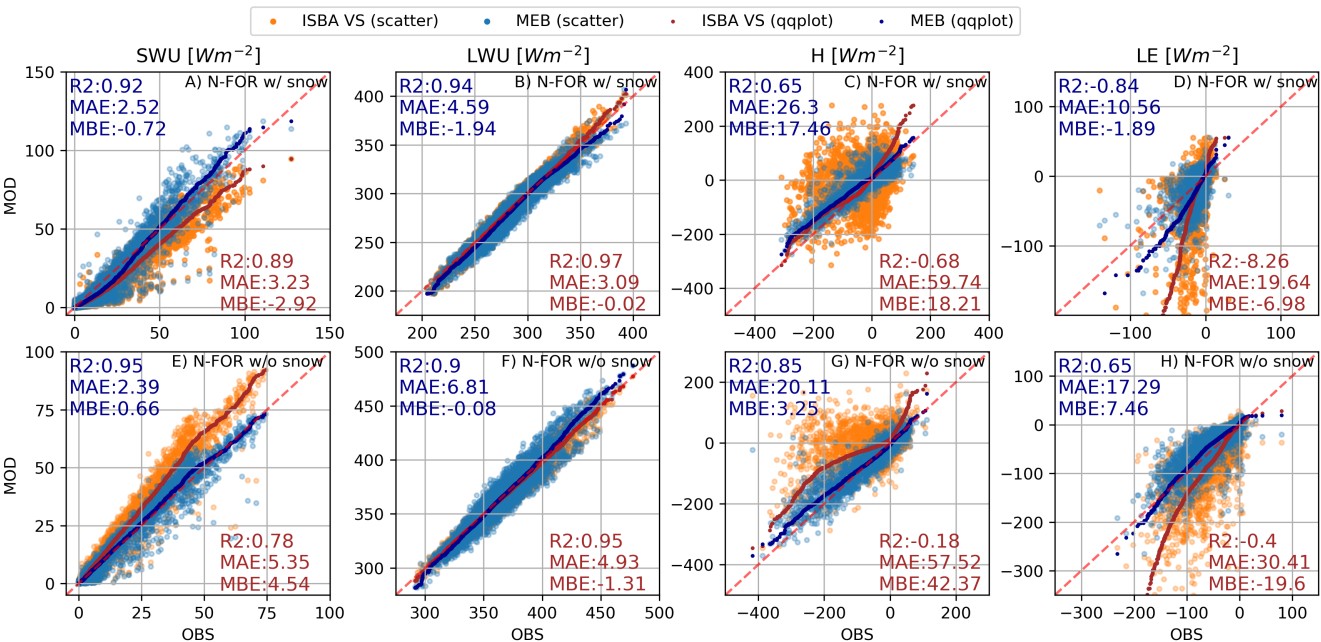

**Figure 11.** Simulated against observed upwards short and longwave radiation (SWU and LWU, columns 1 and 2) and turbulent fluxes (H and LE, columns 3 and 4) on N-FOR site for the full evaluation period. Ground conditions are presented as i) with snow cover (w/ snow, row 1) and ii) without snow cover (w/o snow, row 2).

### 3.3.3  Impact of vegetation representation on surface energy fluxes

Figure 11 compares the deterministic simulations (Sect. 2.5) by MEB and ISBA-VS against observed above-canopy radiative and turbulent fluxes at N-FOR. The snow cover periods are defined according to agreement between MEB simulations and observations, and thus, the ISBA-VS simulations are often snow-free (as seen in Fig. 9).

MEB is superior to ISBA-VS in simulating all energy fluxes. SWU simulations with snow cover are clearly improved by
MEB, but the spread remains relatively large and LSA is underestimated when incoming radiation is small and overestimated when incoming shortwave radiation is higher. The time evolution of LSA on N-FOR and S-FOR is presented in Sect 3.3.4. The LWU is very well simulated by both model configurations. Turbulent fluxes are clearly better simulated by MEB, but the performance metrics of turbulent fluxes (especially LE) are worse than for radiative fluxes. ISBA-VS uses vegetation fraction parameter to scale the partitioning of latent heat flux between vegetation and soil (Eq. 5-6). However, because same roughness
length and turbulent exchange coefficient ($C_H$) is used for both soil and vegetation, the soil evaporation and snow sublimation are likely overestimated and result in clearly wrong partitioning between H and LE (Fig. 11C,D and G,H). In the case of N-FOR, especially the summer energy fluxes were majorly improved by simply assigning the vegetation fraction to unity (full coverage, not shown).



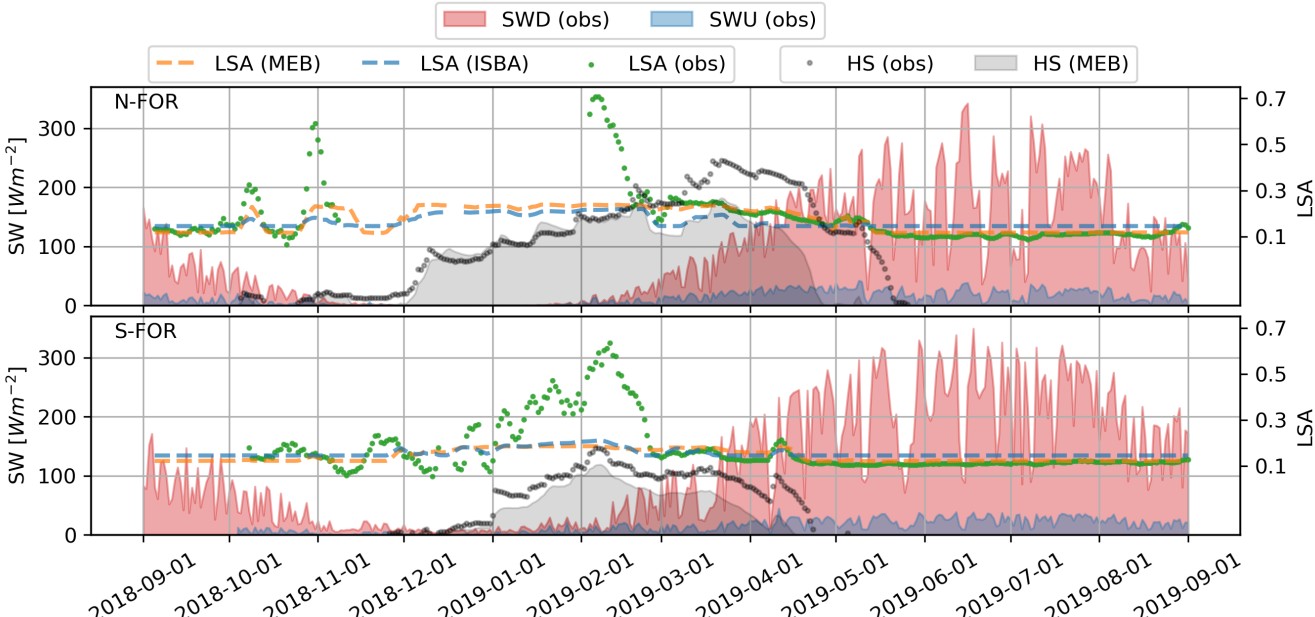

**Figure 12.** Simulated and observed land surface albedo (LSA) and downward and upward shortwave radiation (SWD and SWU) in 2018-2019. LSA simulations and observations are presented as 5-day rolling means. The observed evolution of the height of snow (HS) is not in scale.

### 3.3.4 Evolution of LSA

Figure 12 illustrates the time evolution of modelled and observed LSA and the shortwave components in 2018–2019 on both forest sites. Compared to the measurements, the modelled early and mid winter LSA is underestimated while the spring LSA is slightly overestimated, consistent with results in Sect 3.3.3. The likely reason for winter LSA underestimation is that the models neglect LSA increases due to intercepted snow. The overestimation in spring is presumably due to representing effective LSA of snow and forest canopy with only bulk canopy parameters, as well as effect of spring needle and litter fall decreasing 500 snow albedo.

### 3.4 Summary: Surface energy budget on peatland and forest sites

Finally, to sum up the whole surface energy budget, we compare how the simulated $R_n$ and turbulent fluxes (H+LE) match the observations at the four sites. These deterministic simulations are conducted with simulation setups that provided the best fit to data: the deterministic simulation as M98 in Sect. 2.5 (turbulent exchange as M98 with inclusion of SOC) for the peatland 505 sites, and deterministic MEB simulation (explicit vegetation) for the forest sites. Time series of an example period and scatter plots of all simulations against observations are given in Fig. 13.





Despite the challenges in simulating snow depth evolution at the forest sites, the energy budget simulations are generally better than on the peatlands. Due to the challenges to accurately simulate LSA and surface temperatures on the open sites, the simulated $R_n$ is considerably worse on peatland sites than on forests (see Fig. 7). Especially the high $R_n$, representing the spring conditions, are biased on peatland sites, while the negative $R_n$ (i.e. the winter conditions) are simulated rather well. The challenges in describing forest wintertime LSA and thus SWU (as in Fig. 12 and Fig. 11) does not significantly bias the $R_n$ simulations, as in wintertime the shortwave radiation balance has small role compared to the longwave radiation balance. The results propose that canopy temperature, which particularly in dense forests (e.g. S-FOR) has central role for upward longwave radiation, must be adequately simulated by MEB. When it comes to the turbulent fluxes, the simulations capture the main seasonal patterns. However, there are still high uncertainties (scatter) both on peatland and forest sites. The relative uncertainties in simulated and observed energy fluxes are significantly greater in winter than in summer (compare Fig. 13 vs. Appendix C Fig. C1).

## 4  Discussion

### 4.1  Insights on energy flux partitioning in boreal environments

We used a novel dataset including all surface energy balance components from two peatland and two forest sites. The potential of such data has not yet been fully leveraged in LSM snow model evaluation. Observations showed that in winter the latent heat flux was minimal at all sites and the negative net radiation (down to -50 $\mathrm{Wm}^{-2}$) almost completely counterbalanced by the sensible heat flux (see Fig 13 and Fig. 4). The G had only small contribution to the winter energy budget on the studied peatlands, whereas it has been reported to have a rather important role for the open sites in Canadian Arctic (Lackner et al., 2021), Siberia (Langer et al., 2011) and Svalbard (Langer et al., 2011). This is due to the high heat capacity of the peatland, and its large water storage which progressively freezes from the top keeping the temperatures in the soil-snow surface nearly constant at minimum 0°C. Although the winter average daily $R_n$ and H were similar to those observed in the Canadian Arctic by Lackner et al. (2021), the extremes were considerably larger on the sites studied here. This is most likely due to the more southern location of the study site in Lackner et al. (2021) (56°N), as sites at higher latitudes (Langer et al. (2011) in Siberia 72°N, and Westermann et al. (2009) in Svalbard 78°N) have reported $R_n$ and H extremes closer to those observed in this study.

In the cold regions, periods with stable atmospheric conditions in winter constitute an important part of the annual cycle. Inspired by Lackner et al. (2021), we used the bulk Richardson number to classify atmospheric stability during our study period to unstable ($Ri_b < 0$), weakly stable ($0 \leq Ri_b \geq 0.25$) and strongly stable regimes ($Ri_b > 0.25$) (Table 3). In weakly stable boundary layer, wind shear is sufficient to maintain constant turbulence, while in strongly stable boundary layer weak wind shear leads to weak and intermittent turbulence dominated by drainage winds, wave-turbulence and other mesoscale processes (Steeneveld, 2014; Sun et al., 2012). During snow covered season, the strongly stable turbulence regime was prevailing (78.0 % of time on N-WET and 72.1 % on S-WET), while weakly stable conditions were more rare (14.0 % on N-WET and 6.6 % on S-WET). Regardless of stability regime, we observed considerably higher H than Lackner et al. (2021) at the Canadian





**Figure 13.** Simulated (MOD) against observed (OBS) daily surface energy budget during winter 2018-2019. The left column shows net radiation ($R_n$) and right column presents the sum of turbulent fluxes (H+LE). The scatter plots represent full simulation periods when snow cover was present. The observed evolution of the height of snow (HS) is not in scale.



**Table 3.** Occurrence of different turbulence regimes at S-WET and N-WET. The regimes are defined based on the bulk Richardson number (Ri). Unstable conditions as Ri < 0, weakly stable conditions as $0 \leq$ Ri $\leq 0.25$, and strongly stable conditions as Ri > 0.25.

| | | Turbulence regimes | | |
|---|---|---|---|---|
| Site | Surface | Unstable [%] | Weakly stable [%] | Stable [%] |
| N-WET | all | 32.8 | 14.5 | 52.7 |
| N-WET | snow | 8.0 | 14.0 | 78.0 |
| N-WET | ground | 62.4 | 14.5 | 23.1 |
| S-WET | all | 53.5 | 4.5 | 42.0 |
| S-WET | snow | 21.4 | 6.6 | 72.1 |
| S-WET | ground | 76.7 | 3.5 | 19.8 |

site dominated by weakly stable conditions enhanced by higher winds. We presume this to be due to greater radiative cooling on N-WET and S-WET, that is counterbalanced by large sensible heat flux even under strongly stable conditions.

In spring, increased $R_n$ drives the warming and eventually melting of the snowpack. We observed shorter melting period in the peatlands compared to adjacent forest sites (see in Appendix B Fig. B1). This is in line with Lundquist et al. (2013),

who established that forest tends to increase snow retention relative to open areas in cold climates, while the opposite patterns is typical for warmer climates. Lundquist et al. (2013) postulated that longer snow retention in forest occurs when the effect of shading (slowing down melt) outweighs the impact of longwave radiation enhancement (accelerating melt). Our datasets support this; however, the forest sites tended to also accumulate more snow than the peatland sites (wind erosion is higher on peatlands), which may have contributed to longer snow duration in the forest. Below-canopy measurements of surface fluxes

from snow-covered forest floor would be required to investigate the actual contribution of individual energy fluxes to snow melt, but only a few efforts have been made to acquire such datasets in boreal forest environments (Mazzotti et al., 2020b; Reid et al., 2014). Such data was not available for this study, and hence, only energy fluxes above the canopy were evaluated, as is also commonly done (Napoly et al., 2020; Essery et al., 2009; Rutter et al., 2009).

### 4.2    Implications for simulating snow and energy balance at peatland sites

Our results provide insights and further recommendations on modelling turbulent fluxes over snow. With the ESCROC multiphysics framework, we were able to assess the uncertainties in simulated turbulent fluxes without neglecting the possible contribution from snowpack process descriptions. Our evaluation with multiple years of EC and radiation data of all energy balance components from two subarctic climates allowed deeper analysis of the model performance.

Our simulations showed large differences in surface heat fluxes between turbulent flux parameterizations (Fig. 4) while the fluxes were not impacted as much by alternative snow process parameterizations. Our evaluation evidences that modelling of



turbulent fluxes over snow (i.e. mostly in stable conditions) has major uncertainties, in line with Menard et al. (2021); Conway et al. (2018); Lapo et al. (2019). These uncertainties are larger than in unstable (summer) conditions (Fig. C1), and significantly greater than uncertainty of the radiation balance components. Further, the ESCROC simulations showed that the turbulent ex-

change parameterizations have noticeable impact on snow depth simulations. These results are in line with simulations at Col de Porte, France and ESM-SnowMIP sites (Ménard et al., 2019). In contrast, Lackner et al. (2021) found only small differences between the Crocus turbulence parameterizations in their study in the Canadian Arctic, most likely due to less stable conditions as previously noted.

On peatlands, the M98 (and RI2) option was superior to other stability functions. The improved surface temperature simulations at both sites (absolute biases lower than 0.6°C) provide support to Martin and Lejeune (1998) and Gouttevin et al. (2023), who adjusted the turbulent flux simulations under stable conditions to reproduce surface and air temperature observations. The default turbulent flux parameterization (RIL), although widely used e.g. in NWPs and GCMs (Mahfouf et al., 1995; Salas-Mélia et al., 2005; Voldoire et al., 2013, 2019), provided the poorest fit with the observed surface heat fluxes and snow

depth, and produced a cold bias in snow surface temperature between -1.4°C (S-WET) and -1.3°C (N-WET). This is consistent with ESM-SnowMIP (Menard et al., 2021) results, where the default configuration of Crocus had one of the lowest skill for surface temperature simulations (-2°C mean cold bias) among the compared snow models. However, even with the M98 option, we found rather low skill of turbulent flux simulations. Also Lapo et al. (2019) obtained the best simulations by permitting turbulent exchange under stable conditions (with critical stability threshold) when comparing different stability schemes at a

site in Colorado. Overall, our findings highlight the limitations of MOST theory, to simulate LSM turbulent fluxes under stable atmospheric conditions, and emphasizes the need for further model development and evaluation against observations in various environment.

    Crocus is occasionally used for climate and permafrost studies in the Arctic (Gascon et al., 2014; Sauter and Obleitner,

2015; Graham et al., 2017; Royer et al., 2021), but evaluations of soil temperature profile simulations of this model system in northern peat soils have not been previously made. Decharme et al. (2016) implemented parameterization of SOC in ISBA, and showed that the performance of ISBA coupled to the ES snow scheme improved significantly the soil temperature simulations across northern Eurasia. Our site-level study with Crocus confirms that adequate representation of peat soils (hydraulic and thermal properties, Menberu et al., 2021; Morris et al., 2022; Mustamo et al., 2019) is necessary for accurate simulation of soil

thermal regime and consequent freezing/thawing processes (Dankers et al., 2011; Lawrence and Slater, 2008; Nicolsky et al., 2007). The thermal state and ice/liquid water content have also major cascading effects on runoff generation during snow melt (Ala-Aho et al., 2021).



### 4.3 Implications for simulations at forest sites

#### 4.3.1 ISBA-VS

We showed that turbulent fluxes simulated by ISBA-VS are poorly correlated with the observed ones, consistent with Napoly et al. (2020). We found ISBA-VS to drastically overestimate soil evaporation due to its conceptualization of vegetation and snow cover fraction, resulting in too high LE. This is presumably because ISBA-VS uses same turbulent exchange coefficient ($C_H$) both for computing vegetation evapotranspiration and soil evaporation. At N-FOR, using ISBA-VS with vegetation

fraction set to 1 (i.e. omitting soil evaporation) resulted in significantly improved turbulent flux simulations. In winter, the errors might be also linked to an overestimation of the diurnal amplitude of the ground heat flux from the surface fraction not covered by snow (not shown). Indeed, the snow cover fractions at our forest sites (Eq. 11-13) never exceeded 0.20, meaning that major part of the soil-vegetation composite always remained in direct contact with the atmosphere without the insulating effect of the snow cover.

In terms of LSA, Napoly et al. (2020) found ISBA-VS LSA to depend on forest density: the LSA of dense forest was over-estimated due to overestimation in grid-cell snow covered fraction. Our simulations, in contrast, underestimated the LSA of a sparse forest (N-FOR), which implies a too low snow cover fraction. As demonstrated by Napoly et al. (2020), the snow cover fraction approach of ISBA (Fig. 2B) is essentially a compromise that attempts to retain the insulating impact of the snowpack over the soil while still simulating turbulent exchange from the vegetation. We found this compromise to be largely

biased towards correctly simulated surface energy fluxes at the expense of poor soil temperature simulations, as a major part of the composite was always directly coupled to the atmosphere. The energy exchange between the atmosphere and the soil-vegetation composite directly impacts the snowpack, and leads to strongly biased snow depth simulations, consistent with Napoly et al. (2020). Overall, ISBA-VS with correct tuning (e.g. setting veg. fraction to unity on N-FOR), may be an imperfect but sufficient compromise for forest simulations in applications that foremost require an efficient way to represent grid-cell

averaged surface energy fluxes and are not specifically focused on soil or snow cover state. However, the high sensitivity of such empirically based parameters (e.g. veg. fraction) highlights the limitations of ISBA-VS to provide lower boundary conditions of boreal forests for NWP and GCM applications. Also considering the very low skill obtained in snow depth and soil temperatures for this configuration, its use in hydrological applications or surface offline reanalyses (Le Moigne et al., 2020) is highly questionable. Nevertheless, local-scale evaluation might not directly translate to large scale spatial simulations, as

further discussed in Sect. 4.4.

#### 4.3.2 ISBA-FS

We found that snow and soil simulations in forests were strongly improved when the snow cover fraction was set to unity (ISBA-FS). This adjustments allows the snowpack to fully insulate the soil, similarly to the open peatland sites. The results

suggest that if the focus is on snowpack dynamics and soil temperature simulations, ignoring snow-vegetation interactions is a better compromise than having varying snow cover fraction in the way it is currently implemented in SURFEX (e.g. with only



1 soil column). Consequently, ISBA-FS should be preferred to ISBA-VS in surface reanalyses as in Brun et al. (2013); Vernay et al. (2022). However, ISBA-FS reaches its conceptual limits when forest energy balance, snow, and soil state variables are all of interest. For instance, neglecting snow interception and subsequent canopy snow losses may cause large errors in simulated

SWE in dense forests, and unrealistic contribution of the canopy evapotranspiration may be expected if reanalyses are further used for hydrological modelling. Obviously, highly unrealistic surface energy fluxes would also be expected for any coupling with an atmospheric model.

### 4.3.3  MEB

MEB was developed to solve the aforementioned challenges and reconcile the needs of diverse applications (Boone et al., 2017). It has been previously evaluated on French forest sites, and benchmarked for numerous FLUXNET sites (Napoly et al., 2017). The evaluation of snow-forest interactions has, however, been limited to only three sites in Canada by Napoly et al. (2020) and the ES snow scheme. Our study complements these with two new sites (different vegetation characteristics and climates), and explores MEB performance when it is coupled to the detailed snowpack model Crocus.


Our results show significant improvements in simulated turbulent fluxes and LSA compared to ISBA-VS. However, we could identify two clear systematic biases: LSA was underestimated in winter and overestimated in spring (Fig. 12). The winter LSA was most likely underestimated because intercepted snow increased the LSA, a process assumed negligible in MEB (Napoly et al., 2020). This assumption is based on Pomeroy and Dion (1996), who argued that snow has no significant impact on the

canopy albedo or on $R_n$. Recently, the increase of LSA by intercepted snow has been shown (Webster and Jonas, 2018), and simple descriptions can already be found in some forest snow models (Mazzotti et al., 2020a). Although our results propose the intercepted snow has a clear impact on the LSA, its impact on $R_n$ was negligible. The spring LSA bias is in line with Malle et al. (2021), who found LSA at sparse boreal forests to be overestimated by the LSM CLM5. This could be due to simplistic canopy parameterization of MEB. For instance, different tree species with similar LAI and height have considerably different

geometries and canopies tend to be heterogeneous. In this case, a bulk 'big-leaf' canopy representation may fail to capture complex effect of canopy shading, particularly at low solar elevation angles typical of high latitudes (Malle et al., 2021).

The snow depth simulations by MEB were highly improved compared to ISBA-VS but slightly worse compared to the ISBA-FS, especially at the sparse forest (N-FOR). This suggests that sparse canopies did not majorly alter simulated snow accumulation and ablation, at least when considering snow depths between the trees. Meriö et al. (2023) demonstrated this at

N-FOR with high-resolution UAV snow depth mapping, showing decreased depths at the immediate vicinity of tree trunks, but high snow depth between trees. Although Napoly et al. (2020) found rather good agreement between observed melt out dates and those simulated by MEB, we found MEB to systematically simulate too early snow melt, especially on N-FOR (Fig. 9). These errors are partly explained by inaccuracies in canopy radiative transfer (LSA biases), but they also suggest errors in simulated below-canopy surface heat fluxes; evaluating them would have required complementary observations from below

the canopy. Finally, soil temperatures were better simulated with MEB than with both ISBA-VS and ISBA-FS configurations,



especially at the dense forest (S-FOR). In summary, only MEB with explicit representation of vegetation and ground is able to simulate accurately both snowpack characteristics and soil temperature, as well as the surface energy fluxes.

## 4.4   Limitations and outlook

We forced the model simulations with meteorological data from the study sites, and the data gaps were filled with observations from nearby stations and ERA5 reanalysis product (Hersbach et al., 2020). Intrinsic uncertainties in meteorological observations are known to exist, especially in northern conditions (instrument freezing, snow blocking, undercatch etc., Stuefer et al., 2020). The data gaps further add up possible sources of errors. Uncertainties in model forcing can affect model-data comparisons, especially during the gap-filled periods (Raleigh et al., 2015). Our EC-based fluxes are among the longest dataset ever

used for the evaluation of turbulent flux simulations over snow. The EC-data, however, contains both random and systematic uncertainties (e.g. Aubinet et al., 2012). The absolute values of winter H and LE are small in northern conditions, and their relative uncertainty is high; compared to summertime measurements the wintertime energy balance closure ratio is typically poorer (Reba et al., 2009; Molotch et al., 2009; Launiainen, 2010). As our analysis uses numerous site years from multiple sites, and we used established quality criteria for filtering the EC-fluxes, we expect uncertainties in flux data do not significantly

affect the study results. Moreover, the conclusions regarding the validity of each model version were not affected by selected quality flag (Sect. 2.4.2).

Some potentially important snow processes on subarctic sites are still absent in Crocus. These include wind-induced erosion and accumulation due to snow transport and internal water vapor transfer due to large temperature gradient in the snowpack.

Wind-induced snow transport can move mass laterally and change the properties of snow (Pomeroy and Essery, 1999; Meriö et al., 2023; Liston and Sturm, 2002), and is especially noticeable on open peatlands. In Crocus, wind modifies the properties of falling snow (Vionnet et al., 2012) but without any lateral transport or modifications of the mass. Although we achieved satisfactory model performance even without accounting for this process, Meriö et al. (2023) showed notable wind transport in transition zones between open peatland and forest at the N-WET site, that may alter the total snow mass and the properties of

the surface snow layer. Although the spatial scale of wind transport prevents an explicit simulation of this process in large scale LSMs, improved parameterizations of the wind impact of near-surface snow properties should be considered in the future.

Then, the lack of internal water vapor transfer by diffusion and/or convection in the snowpack has been suspected to be responsible for errors in simulated snow properties (density, microstructure) in Arctic snowpack (Barrere et al., 2017; Domine et al., 2018) and consequently in thermal conductivity and soil thermal regime. Complementary observations and model eval-

uation are required to understand if the simulated snow properties are also affected by this kind of errors in our study cases. Furthermore, the spring and autumn conditions on the peatlands are difficult to correctly simulate; in addition to the snow cover, also e.g. ponding of liquid water and refreezing of the ponds are not uncommon (Noor et al., 2022) and can alter the LSA. These processes are included neither in ISBA nor Crocus.





In forests, the spatial heterogeneity of snow cover can be high, as demonstrated by numerous studies (Marttila et al., 2021; Mazzotti et al., 2020b; Noor et al., 2022) and confirmed by our data (Fig. 9). The small-scale forest structure has an important role in the evolution of the snow cover, and may affect the representativeness of point measurements (Bouchard et al., 2022). Consequently, the comparison of point observations and models intended for forest stand and larger scales (such as the big-leaf approach of MEB), can be flawed (Essery et al., 2009; Rutter et al., 2009). The forests considered in this study were

rather homogeneous (Aurela et al., 2015; Hari et al., 2013) and our EC-data can be assumed to capture the footprint average fluxes. Some attempts to capture the spatiotemporal variability of below-canopy energy fluxes, representing the forest floor and understory, have recently been made with distributed measurements or moving platforms (Malle et al., 2019; Mazzotti et al., 2019), yet these datasets are short-term. In particular, below-canopy measurements of turbulent energy exchange are scarce and have to date not been routinely used in snow modeling (e.g. Launiainen, 2010; Launiainen et al., 2005; Molotch et al.,

2009; Marks et al., 2008)). Simultaneous above- and below-canopy measurements may have great potential for snow model evaluations at forest sites. In the absence of energy flux measurements below the canopy, observations of soil temperature and snow conditions allowed an indirect assessment of below-canopy energy budget, and highlighted necessary improvements. In the future, more realistic below-canopy and above-canopy heat flux simulation could be achieved by more sophisticated canopy representations, including multiple layers and species (e.g. Bonan et al., 2021; McGowan et al., 2017; Launiainen et al., 2015;

Gouttevin et al., 2015). For site-level or limited area modelling, high resolution models that explicitly resolve tree-scale canopy structure are a promising alternative to traditional LSMs (Broxton et al., 2015; Mazzotti et al., 2020b).

## 5    Conclusions

We used eddy-covariance based energy flux data, radiation balance and snow depth and soil temperature measurements in two

boreal and subarctic peatlands and forests to evaluate turbulent exchange parameterizations and alternative approaches to represent soil and vegetation in LSMs. While our model experiments relied on the SURFEX platform, our findings are transferable to other model systems. Our evaluation with the ESCROC ensemble Crocus snow model framework gives confidence that uncertainties in snow processes not evaluated in this study do not affect the robustness of our main conclusions summarized below.

Our peatland simulations showed that using a stability correction function that increases the turbulent exchange under stable atmospheric conditions is imperative to simulate the snow energy budget and to capture snow melt events driven by the turbulent fluxes. Although this adjustment led to major improvements under stable conditions during snow cover, the model performance remained lower than those simulated under snow-free conditions. Furthermore, correct hydraulic and thermal parameterization of the peat soils was found necessary to reproduce the observed soil thermal regime. The findings have direct implications for

modelling snow dynamics, peatland hydrology as well as permafrost dynamics.

Our forest simulations showed that the surface energy budgets were well simulated by the explicit big-leaf approach (MEB), while the composite soil-vegetation approach (ISBA-VS) performance was only satisfactory after an adjustment of a sen-

sitive vegetation fraction parameter. In particular, shortwave and longwave radiation balances were simulated well by both approaches, whereas the turbulent fluxes had significantly higher uncertainty. Only the explicit vegetation model (MEB) was

able to simultaneously simulate realistic surface energy budget and snow/soil conditions, while the composite approaches succeeded in either simulating the correct surface energy budget (ISBA-VS) or snow/soil conditions (ISBA-FS) depending on the configuration.

The generality of our findings should be tested by additional snow model and LSM evaluation studies, extended to more

contrasting climates and different ecosystem types. For this purpose, reference evaluation datasets should be complemented with more boreal and Arctic sites and observations of all components of surface energy balances, particularly turbulent fluxes. Such a dataset would facilitate similar experiments with other models.

With well-selected model configuration and parameterization, SURFEX model platform can realistically simulate surface

energy fluxes and snow and soil conditions in the subarctic and boreal peatlands and forests. The common version of ISBA (ISBA-VS) can provide rather realistic lower boundary conditions for numerical weather prediction (NWP) and global circulation models (GCMs), in expense of non-realistic predictions of forest snow and soil conditions necessary for hydrological applications. We expect that the future inclusion of MEB in operational systems will reconcile these applications by improving the skill of weather forecasts, climate scenario simulations and hydrological forecasts. Our results can be used to inform the

choice of model configuration for studies of subarctic and boreal regions ecology, hydrology and biogeochemistry under the ongoing environmental change.

*Code availability.*

The SURFEX is an open source project (http://www.umr-cnrm.fr/surfex) but it requires registration. The full procedure

and instructions are available at https://opensource.umr-cnrm.fr/projects/snowtools_git/wiki/Procedure_for_new_users. The SURFEX version used in this work is available in git (tagged as *boreal_ecosystems*). The ESCROC version developed for external SURFEX users, is available at https://github.com/bertrandcz/CrocO_toolbox.

*Data availability.*

Data are available upon request from the authors.





## 755 Appendix A: Model evaluation data availability

The availability periods of the model evaluation data are presented in A1.

**Table A1.** Model evaluation data availability for each site.

| Variable | N-WET | S-WET | N-FOR | S-FOR |
|---|---|---|---|---|
| Height of snow | 2017-11–2021-05 | 2016-09–2021-07 | 2013-09–2020-09 | 2008-09–2021-07 |
| Soil temperature | 2013-09–2019-12 | 2017-06–2021-07 | 2016-09–2020-09 | 2008-09–2021-07 |
| Outgoing LW flux | 2017-07–2021-06 | 2016-09–2021-07 | 2013-09–2021-07 | 2008-09–2021-07 |
| Outgoing SW flux | 2017-07–2021-06 | 2016-09–2021-07 | 2013-09–2021-07 | 2008-09–2021-07 |
| Sensible heat flux | 2013-09–2021-06 | 2016-09–2020-12 | 2013-09–2021-07 | 2008-09–2021-07 |
| Latent heat flux | 2013-09–2021-06 | 2016-09–2020-12 | 2013-09–2021-07 | 2008-09–2021-07 |
| Ground heat flux | 2013-09–2017-07 | 2016-09–2021-07 | 2013-09–2021-02 | 2008-09–2021-07 |



**Appendix B:  Comparison of observed snow depths in the forest and peatland sites**

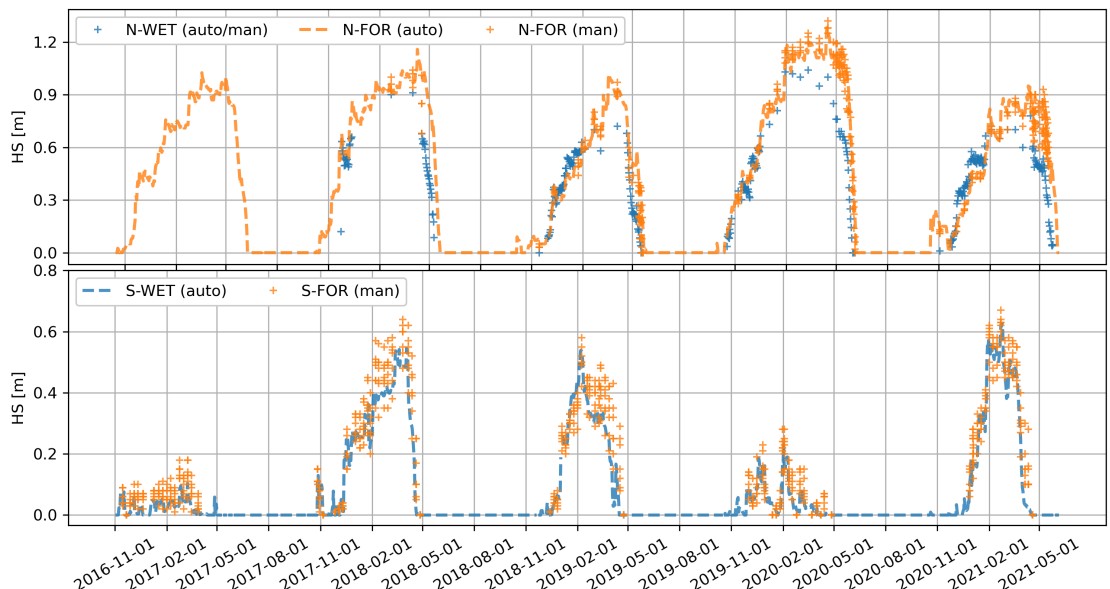

**Figure B1.** Height of snow observations in forest sites vs. nearby peatland sites



## Appendix C: Surface energy budget during summer



**Figure C1.** Simulated (MOD) against observed (OBS) daily surface energy budget during summer 2018-2019 (center column). Left column shows net radiation ($R_n$) and right column the sum of sensible (H) and latent (LE) heat fluxes. The scatter plots represent full simulation periods with snow free conditions.





## Appendix D: Effect of snow cover fraction parameter $w_{sw}$ on simulated snow depths

**Table D1.** Summary of snow cover fraction and values of $w_{sw}$ used in different SURFEX/ISBA applications. The parameter $w_{sw}$ is rarely documented, and hence, these application specific values were obtained through communications with the authors.

| Application | Name | Domain | Resolution | Snow fraction | $w_{sw}$ | Reference |
|---|---|---|---|---|---|---|
| Numerical weather prediction | AROME, ARPEGE | Europe (many) | 1.3 - 10 km | varying | 5 | Bengtsson et al. (2017) Courtier et al. (1991) |
| Global climate modelling | CNRM-CM6 | Global | 100 km | varying | 2 | Decharme et al. (2019) |
| Regional climate modelling | CNRM-AROME | European Alps | 2.5 km | varying | 1 | Caillaud et al. (2021) |
| Regional climate modelling | CNRM-ALADIN | Europe, North Africa | 12 km | varying | 2 | Nabat et al. (2020) |
| Hydrological modelling | SIM2 | France | 8 km | varying | 0.2 | Le Moigne et al. (2020) |
| Regional reanalysis | CERRA-Land | Europe | 5.5 km | varying | 0.1 | Verrelle et al. (2021) |
| Snow cover reanalysis | S2M | French Alps | massif-scale | full (1) | - | Vernay et al. (2022) |
| Snow cover reanalysis | ERA-Interim-Crocus | Northern Eurasia | 80 km | full (1) | - | Brun et al. (2013) |
| Avalanche hazard forecasting | S2M | French Alps | massif-scale | full (1) | - | Morin et al. (2020) |

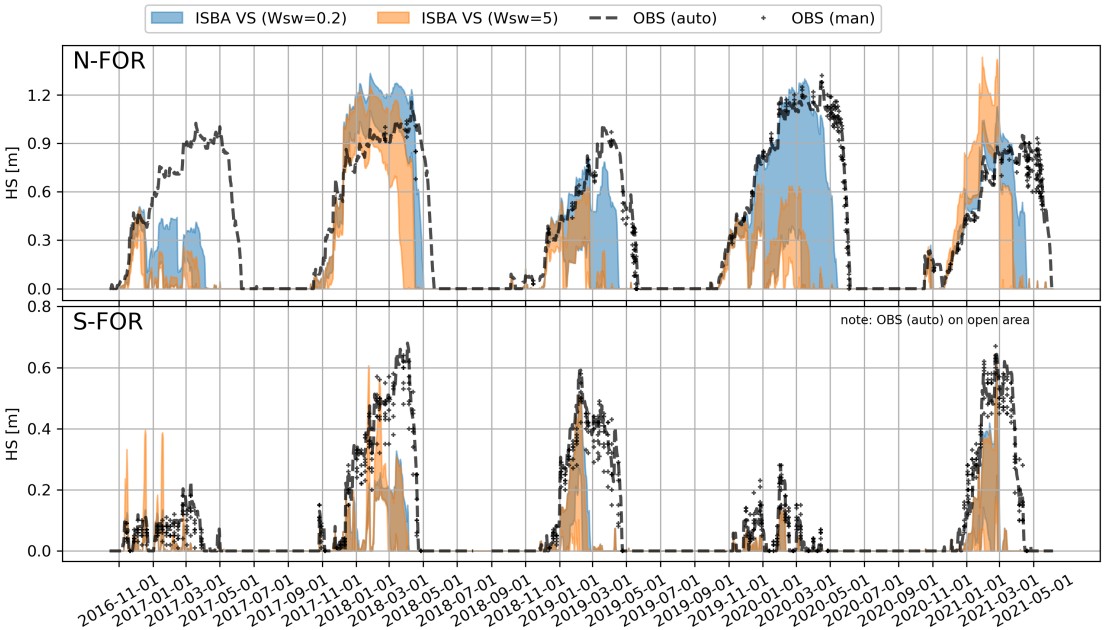

**Figure D1.** Effect of $w_{sw}$ parameter on snow depths simulated by ISBA-VS. The envelopes visualize the corresponding ensemble spreads between minimum and maximum values.



*Author contributions.*

JPN, ML, SL, PA and HM designed the research. JPN led the study and was in charge of the experiments. JPN, GM and ML performed the model evaluations with scientific contributions of SL, HM and PA. JPN was responsible for writing the article, with significant contributions from GM, ML and SL and input from all the authors. BC and MF supported the ensemble simulations and other technical aspects. MA, AL, PK, PA and HM provided surface energy data and ancillary data.

*Competing interests.*

The authors declare that they have no conflict of interest.

*Acknowledgements.* This work was funded by the Academy of Finland (ArcI Profi 4). GM was funded by the Swiss National Science Foundation (Grant P500PN_202741). SL acknowledges GreenFeedBack project from the EU Horizon Europe – Framework Programme for Research and Innovation (no. 101056921). PA was funded by the Academy of Finland Research Fellow grant (no. 347348). PK, MA and AL thank the support from the ACCC Flagship funded by the Academy of Finland (no. 337549 and no. 337552) and ICOS-Finland by University of Helsinki and the Ministry of Transport and Communication. MA and AL also acknowledge the Academy of Finland (UPFORMET, grant no. 308511). The authors would like to thank Bertrand Decharme, Christine Delire, Bertrand Bonan at CNRM/GAME and Marie Dumont at CNRM/CEN for their great support during this work. CNRM/CEN is part of Labex OSUG@2020.



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
