# Peer review of "Modelling snowpack dynamics and surface energy budget in boreal and subarctic peatlands and forests"

_EGUsphere, 2023_

## Referee Comment (RC1)

This study evaluated model parameterizations and configurations determining turbulent exchange and soil and vegetation and their influences on snowpack and surface energy fluxes using station-based eddy-covariance based energy flux data, snow depth, and soil temperature observations in two boreal and subarctic peatlands and forests in Finland. They found that a stability correction function is a key part to simulate sensible and latent heat fluxes over snow. Also, a realistic soil texture (soil organic carbon) parameterization led to improvement of the soil temperature simulations in peatlands.

The study presents insightful results describing model configuration in boreal peatlands and forests where are the most challenging regions to estimate snow energy balance budget. While the findings from this study should be tested by other snow model and LSMs to be widely generalized, as a case study, this study provides useful implications for choosing suitable turbulent flux parameterization and model structures to better estimate snow energy fluxes in soil-vegetation-snow interactions.

Even though I feel this manuscript is lengthy as a single manuscript, the paper is generally written well, and the presentation quality of the figures is great. In my opinion, however, concerns need to be addressed upon before publication is warranted. I have given a few suggestions below that may contribute to the improvement of the manuscript.

**General comments**

1. It is unclear to me about differences in MINERAL vs. SOC model runs to assess for assessing soil thermal regimes. Is the two model runs' difference soil texture only? In Table 2 of peatland sites, the authors assumed 100% SOC of soils. But in Figure 8, MINERAL was run as well. Does it mean the authors used "mineral soil" instead of 100% SOC with the same model? I don't know clearly how the ISBA parametrize soil texture but if "mineral soil" is continuous values (not categorized) within the model, it would be much more useful if you provide sensitivity of soil temperature and energy fluxes to different SOC percentages, e.g., 0, 20, 40, 60, & 100%).
2. Even though the MINERAL vs. SOC comparison showed a more realistic soil texture (SOC 100%) provided a better performance, there were still unexplained portions of the modeled soil temperature as compared to observations. I would recommend including some discussions regarding the unexplained portions – how the model could additionally improve soil thermal regimes (e.g., soil moisture contents).
3. I found "not shown" in several places. I understand the authors may want to keep the manuscript's length readable. However, "not shown" may not be helpful for potential readers who may want to keenly read this paper. I would recommend including any additional figures and tables that can support the findings from this study as Supporting Information (or Appendix).
4. I wonder if there are any specific reasons focusing on 2018-19 winter. Was the winter a representative year in terms of snow climatology (e.g., moderate snow condition)? I would suggest providing a justification regarding that.

**Detailed comments**

Figure 1. I think the current version of the figure does not extremely useful. Any detailed photos at the sites with positions of flux towers and soil temp & snow depth sensors may be needed. That would be helpful to better understand observations for those who are not familiar with those

environments. Additionally, please include the site points on aerial photos with a scale bar. If the authors need more space, I recommend including them into Appendix.

L160 transpiration from the vegetation, Etr

L209 It would be good to include a citation (Vionnet et al., 2012) here

L222-223 My understanding is that at the bottom of the snowpack, Crocus is coupled to the soil components of the ISBA LSM. Please describe it explicitly.
What does "a mass and energy-conserving semi-implicit solution" mean? A detailed description of this would be helpful to understand.

L223 "Is there a specific equation of the heat conduction G related to the temperature gradient? I would suggest including this or citing a relevant reference.
Did you use the heat conduction G as the same concept to the surface heat flux into the soil-vegetation composite (G; Eq 1)? If not, please use a different acronym.

L291 What does "100% of SOC" mean? How much is this when it's converted as kg/m2? In Table 2, this should be expressed as kg/m2 for consistency.

Regarding the assumption, I would strongly suggest providing any justification of this assumption with sufficient references (e.g. why this is reasonable).

L360 For model evaluation, have you considered daily-maximum and/or minimum values in addition to the daily-average values particularly for surface energy flux and soil temperature? Because of strong diurnal patterns of these variables, I think it would be useful.

Figure 7 please change the color of HS (obs) to black or others to make a difference from HS (mod).

L425 Should "N-FOR" be corrected as "N-WET"? Figure 7 doesn't provide N-FOR time series

L426-426 I believe the authors used the ISBA-VS approach only on the peatland sites. But I'm curious how the results will be changed with ISBA-VS because this statement indicates that the uncertainty would be improved when the model considers fractional snow cover. Please consider

In Figure 7, during the melting period, there were different model errors in LSA between two sites (e.g., N-WET: underestimation but S-WET: overestimation). Please include some discussion regarding this.

L441 Please reword this sentence. It's unclear to me.

L443-447 The authors mentioned "On both sites" but a figure of N-WET wasn't presented here. I would recommend including the N-WET results as well (even in Supplementary Info) for those who are interested in. I'm not a big fan of "not shown".

Did "SOC" simulation mean "M98" simulation (L345-347)? Because the acronym SOC also means "soil organic carbon" itself, it's a little bit confusing. Please use it for either one. Also clarify what the two simulations are fundamentally different in Sec 2.5. To my understanding, in MINERAL, the soil was characterized as "mineral" while SOC (or M98) as "100% SOC". Is this correct?

L456 I don't think Figure 9 shows the ISBA-VS model version heavily overestimates accumulation, except 2021 for N-FOR.

L456-458 Can you provide a figure comparing two runs with between Wsw = 5 and 0.2? What does it physically mean? In section 2.3.1, the author mentioned "The coefficient Wsw relates to vegetation characteristics" but I don't think it would be enough to physically understand the parameterization and the results here. Please provide more details.

L462-463 How did the authors know that the results are due to overestimated compaction and uncertainties in snowmelt? For example, have you compared snow density (as a proxy of compaction) with observations? If so, why did the errors of compaction and melt processes occur in terms of model physics - particularly snow energy processes?

L464 Again, please include your relevant result in Supporting Info instead of "not shown".

I think the authors would be able to discuss more in this section (3.3.2) because there are lots of interesting patterns such as the cold biases in fall. Can you add air temperature in this figure for comparison purposes? I'm guessing that the modeled soil temp seems to be much more sensitive to air temperature. Also, I would suggest adding discussion about the differences in soil temperature between ISBA-FS and MEB.

L489-491 So how would the roughness length and turbulent exchange coefficient (CH) be used differently for soil and vegetation to minimize the overestimations of soil evaporation and sublimation?

Figure 12, for N-FOR, were the LSA observations missing or larger than 0.7? Please justify. To me, it is questionable why the LSAs from models were constant throughout the season, likely regardless of the existence of snowpack, which don't seem to make sense. Please discuss the potential causes.

L498 "neglect" -> "do not estimate"?

L547-549 I don't think this statement is sufficiently supported by this study because the snow depth patterns from the models highly depend on year-by-year (Figures 5 & 9). "Wind erosion is higher in peatlands" is generally correct, but have you seen the wind data in the sites to see any wind effect?

Table 3 "Weakly stable" sounds somewhat weird. Please consider an alternative.

L584-587 Does Crocus itself provide soil temperature simulation? I believe it only estimates snow and is coupled with ISBA to estimate soil parts. Thus it would be better to state something like "ISBA coupled with Crocus"

L597 I don't think the authors explicitly showed soil evaporation from the results. Please rephrase it.

L717-718 Please reword.

---

## Author Comment (AC1)

This study evaluated model parameterizations and configurations determining turbulent exchange and soil and vegetation and their influences on snowpack and surface energy fluxes using station-based eddy-covariance based energy flux data, snow depth, and soil temperature observations in two boreal and subarctic peatlands and forests in Finland. They found that a stability correction function is a key part to simulate sensible and latent heat fluxes over snow. Also, a realistic soil texture (soil organic carbon) parameterization led to improvement of the soil temperature simulations in peatlands.

The study presents insightful results describing model configuration in boreal peatlands and forests where are the most challenging regions to estimate snow energy balance budget. While the findings from this study should be tested by other snow model and LSMs to be widely generalized, as a case study, this study provides useful implications for choosing suitable turbulent flux parameterization and model structures to better estimate snow energy fluxes in soil-vegetation-snow interactions.

Even though I feel this manuscript is lengthy as a single manuscript, the paper is generally written well, and the presentation quality of the figures is great. In my opinion, however, concerns need to be addressed upon before publication is warranted. I have given a few suggestions below that may contribute to the improvement of the manuscript.

*We would like to thank the Reviewer for the throughout assessment and positive comments regarding our study. We are happy that the Reviewer sees our results as insightful and the article as useful contribution to snow and LSM modelling community.*

*We would like to point out and apologize for two mistakes found in the data processing during an additional quality check motivated by the reviews. Thus, we would like to thank the reviewers for boosting our data quality control process.*

*First, we found 2h discrepancy in time zones between different data sources, notably between the main observations used for forcing and the data used for gap-filling. We have converted all the forcing and evaluation data into UTC time zone and ran new simulations. We will update the figures to include the corrected simulations in the revised manuscript. These changes do not have any major impact on the results of the study.*

*The second error was in the wind forcing reference height used for the S-WET site. The S-WET wind reference height had been assigned as in the contiguous S-FOR site (16.8 m), whereas in reality wind was measured at 3 m height. This means that simulated wind speed near the surface of the low-vegetated S-WET site was unrealistically low in our simulations. We have found that this reduces the sensitivity of turbulent fluxes (mainly sensible heat exchange) simulations to different turbulent exchange parameterizations, as the site experiences stable conditions less often than previously simulated. This correction does not change the results showing that M98 turbulence option is superior to classic stability correction function (RIL). However, this correction does change the snow depth simulations on S-WET. Figure 5. changes in a way that all the turbulent exchange parameterizations succeed in simulating the snow melt (see the revised Fig. 5 below). Results related to these changes will be corrected in the revised manuscript.*

[Figure]

*Revised Figure 5. Time series of snow depths simulated by ISBA-ESCROC. The 35 ensemble members are grouped by their turbulent flux parameterization, and the spread of each group is presented in colored ranges. Observed snow depths are presented in black dots and dashed lines.*

*Below we provide our detailed replies to general and detailed comments.*

**General comments**

1.  It is unclear to me about differences in MINERAL vs. SOC model runs to assess for assessing soil thermal regimes. Is the two model runs' difference soil texture only? In Table 2 of peatland sites, the authors assumed 100% SOC of soils. But in Figure 8, MINERAL was run as well. Does it mean the authors used "mineral soil" instead of 100% SOC with the same model? I don't know clearly how the ISBA parametrize soil texture but if "mineral soil" is continuous values (not categorized) within the model, it would be much more useful if you provide sensitivity of soil temperature and energy fluxes to different SOC percentages, e.g., 0, 20, 40, 60, & 100%).

*This is a good point; the descriptions of peatland 'MINERAL' simulations were not addressed with sufficient detail and will be improved in the revised manuscript. These two simulations differ only in the soil texture. For the peatland 'MINERAL' simulation, we parameterized the soil texture (sand and clay fractions) identically as had been measured and used for the contiguous forest site (see Table 2.).*

*We are going to revise the Table 2. to be clear that the clay and sand fractions for peatlands are only effective below 1 m depth. The text at L340-L347 we are going to revise as:*
*"For a more detailed evaluation of two contrasting turbulent exchange options within ESCROC, we conducted deterministic (ISBA-FS) simulations with i) site parameters as shown in Table 2. and all default ESCROC parameterization as in Fig 2. in Lafaysse et al. (2017) (processes listed in Sect. 2.3.2, referred as RIL), and ii) site parameters as shown in Table 2. and all the default ESCROC parameterizations except the turbulent exchange option switched to M98 (referred as M98).*

*Moreover, we explore the influence of soil texture on the soil thermal regime and on snowpack dynamics. Hence, an additional deterministic simulation was conducted where the soil was characterized as mineral soil, identical as had been measured and used for the contiguous forest site, while turbulent exchange was set to M98 (referred as MINERAL). The MINERAL simulation was compared to the previously described M98 simulation, where the soil was characterized as fully organic until 1 m depth  (Table 2)."*

*Rather than assessing uncertainties in quantifying the soil organic matter, the purpose of this sensitivity test was to show the consequences if the soil organic matter is neglected on boreal peatlands, which is the case in many applications of the SURFEX LSM. In addition, the content of organic matter in such boreal peatlands can be very close to 100%. For instance, Väliranta and Mathjissen (2021) measured mean loss on ignition value 96,6 % in the first meter of the peat profile at S-WET. Further, Muhic et al. (2023) also measured organic matter to be nearly 100% for the first 40 cm depth at a location just next to the N-WET peatland (see Muhic et al. 2023 Fig. 1b "FP2"). We are going to revise the manuscript by adding these citations to justify the use of 100% organic matter for model parametrization.*

2. Even though the MINERAL vs. SOC comparison showed a more realistic soil texture (SOC 100%) provided a better performance, there were still unexplained portions of the modeled soil temperature as compared to observations. I would recommend including some discussions regarding the unexplained portions – how the model could additionally improve soil thermal regimes (e.g., soil moisture contents).

*Indeed, the soil temperature simulations were still far from perfect and could be explained by some missing processes in the model. The missing processes were already discussed for snow cover, but we acknowledge discussion was not complete for potentially missing or not-well represented soil processes. In the revised manuscript, we will add a discussion about uncertainties in soil moisture content, which is especially important for the peatland as the moisture is expected to be controlled by water table dynamics, potentially affected by lateral flows not accounted for by this 1-dimensional LSM approach. Another source of uncertainty originates from the soil-vegetation composite approach where the vegetation layer is not explicitly represented above the soil. We will comment on this in the discussion of the revised manuscript as well.*

3. I found "not shown" in several places. I understand the authors may want to keep the manuscript's length readable. However, "not shown" may not be helpful for potential readers who may want to keenly read this paper. I would recommend including any additional figures and tables that can support the findings from this study as Supporting Information (or Appendix).

*Thanks for this comment, indeed we tried to keep the manuscript length readable by not showing additional figures. We have gone through our 'not shown' results and our plan is following:*

- *Fig. 8: S-WET height of snow plot will be moved to Supplement. We will add previously 'not shown' N-WET height of snow to the Supplement plot as well.*
- *L464: Comparison of N-FOR simulated and observed SWE will be added to Supplement*
- *L493: "In the case of N-FOR, especially the summer energy fluxes were majorly improved by simply assigning the vegetation fraction to unity (full coverage, not shown)." This plot will also be added to Supplement*
- *L602: We believe 'not shown' is not necessary here and will be removed*

4. I wonder if there are any specific reasons focusing on 2018-19 winter. Was the winter a representative year in terms of snow climatology (e.g., moderate snow condition)? I would suggest providing a justification regarding that.

*We chose this season as it contains the best coverage of energy flux data (least gaps) for all the sites, and it also represents well the typical snow conditions on the sites. We will clarify this in the revised manuscript.*

**Detailed comments**

Figure 1. I think the current version of the figure does not extremely useful. Any detailed photos at the sites with positions of flux towers and soil temp & snow depth sensors may be needed. That would be helpful to better understand observations for those who are not familiar with those environments. Additionally, please include the site points on aerial photos with a scale bar. If the authors need more space, I recommend including them into Appendix.

*Thank you, we have revised Fig 1. accordingly (see revised figure at the end), and it will be added to the revised manuscript. We are also going to add site photos in Supplement (see at the end).*

L160 transpiration from the vegetation, Etr

*Thanks for pointing out, this will be corrected.*

L209 It would be good to include a citation (Vionnet et al., 2012) here

*The citation will be included in the revised manuscript.*

L222-223 My understanding is that at the bottom of the snowpack, Crocus is coupled to the soil components of the ISBA LSM. Please describe it explicitly.

What does "a mass and energy-conserving semi-implicit solution" mean? A detailed description of this would be helpful to understand.

*As in many land surface models, the snow and soil modules in SURFEX are separated modules in which the thermal diffusion is first solved for snow and then for soil during the same time step. However, as the ground-snow heat flux is necessary at both steps, the thermal diffusion in snow is solved by using the first ground layer temperature from the previous time step and assuming it remains constant. The resulting snow-soil heat flux is then transmitted to the soil scheme to solve the heat diffusion in soil with a flux boundary condition, thus conserving energy in the coupling. Beyond this treatment of the soil-snow interface, the heat diffusion is solved in both snow and soil components with an Euler backward difference implicit numerical scheme (i.e. expressing snow/soil temperatures at the end of the time step in temperature evolution equations). We are going to summarize this in the revised manuscript at L221-223:*

*"The bottom of the snowpack and the uppermost soil layer are fully coupled with a mass and energy-conserving semi-implicit solution. The semi-implicit solution refers to a coupled system in which both components are solved separately with an implicit approach considering that the state of the second system remains constant during the solving of the first system."*

L223 "Is there a specific equation of the heat conduction G related to the temperature gradient? I would suggest including this or citing a relevant reference.

*The heat conduction flux is expressed using Fourier equation relating G to vertical temperature gradient and thermal conductivity as described in Eq. 4 of Decharme et al., 2011. We will add the reference to this Equation in the revised manuscript. The thermal conductivity and heat capacity are described using pedo-transfer functions (Noilhan and Mahfouf, 1996 and Peters-Lidard et al., 1998). These were already cited in the initial manuscript but will be introduced immediately after the introduction of heat conduction in the revised manuscript.*

Did you use the heat conduction G as the same concept to the surface heat flux into the soil-vegetation composite (G; Eq 1)? If not, please use a different acronym.

*G represents the total heat flux between two model components. In Eq. 1, it refers to the total heat flux between surface and atmosphere, while in Eq. 8, different subscripts indicate that it can represent the flux between the two first soil layers, two first snow layers, or between soil and snow. In the revised manuscript, all these terms will be more accurately defined and the subscript 0 will be added in Eq. 1 to better identify its meaning.*

L291 What does "100% of SOC" mean? How much is this when it's converted as kg/m2? In Table 2, this should be expressed as kg/m2 for consistency.

*We understand our definition of 100 % SOC was confusing. We meant that the soil is fully organic matter without any fraction of sand or clay. The 100% organic matter corresponds to 93,5 kg/m.$^2$ We are going to revise our terminology and Table 2. accordingly.*

Regarding the assumption, I would strongly suggest providing any justification of this assumption with sufficient references (e.g. why this is reasonable).

*Please see our answer to this in the beginning of general comments. We are going to add justification in the revised manuscript.*

L360 For model evaluation, have you considered daily-maximum and/or minimum values in addition to the daily-average values particularly for surface energy flux and soil temperature? Because of strong diurnal patterns of these variables, I think it would be useful.

*Yes, we have considered maximum and minimum values as well but time series become quickly messy with too much information. The diurnal patterns are strong, and they are included in the scatterplot comparison of 6-hour averages of energy flux observations and simulations. See example Fig. 6, Fig. 7, Fig. 11 and Fig. 13.*

Figure 7 please change the color of HS (obs) to black or others to make a difference from HS (mod).

*The figure will be adapted as suggested*

L425 Should "N-FOR" be corrected as "N-WET"? Figure 7 doesn't provide N-FOR time series

*Thanks for pointing out this error which will be corrected in the revised manuscript*

L426-426 I believe the authors used the ISBA-VS approach only on the peatland sites. But I'm curious how the results will be changed with ISBA-VS because this statement indicates that the uncertainty would be improved when the model considers fractional snow cover. Please consider

*You probably mean that we only used ISBA-FS on peatlands. Using ISBA-VS in peatland sites would be less sensitive than on forest sites as the vegetation height is much lower, and therefore the snow fraction is quickly closer to 1 (i.e. identical to the ISBA-FS configuration). In any case, considering the interactions between snow and vegetation on peatlands would be likely improved using MEB as well*

*for low vegetation. However, as MEB has never been applied for snow-covered low vegetation, complementary developments and evaluations are required and this would be beyond the scope of our work. In-progress works in CNRM intend to generalize the applicability of MEB for these conditions. This point will be discussed in the revised manuscript.*

In Figure 7, during the melting period, there were different model errors in LSA between two sites (e.g., N-WET: underestimation but S-WET: overestimation). Please include some discussion regarding this.

*In fact, the overestimation on S-WET is not during melt. This small overestimation is between the two main melt events, after a very light snowfall. We are going to add this in the revised manuscript.*

L441 Please reword this sentence. It's unclear to me.

*We will modify this to the revised manuscript as:*

*"This means soil temperature variations are attenuated in M98 (including SOC) compared to MINERAL, and  this  attenuation becomes increasingly important in deeper soil layers (Fig. 8)."*

L443-447 The authors mentioned "On both sites" but a figure of N-WET wasn't presented here. I would recommend including the N-WET results as well (even in Supplementary Info) for those who are interested in. I'm not a big fan of "not shown".

*We are going to add similar figure concerning N-WET soil temperatures in the Supplementary.*

Did "SOC" simulation mean "M98" simulation (L345-347)? Because the acronym SOC also means "soil organic carbon" itself, it's a little bit confusing. Please use it for either one. Also clarify what the two simulations are fundamentally different in Sec 2.5. To my understanding, in MINERAL, the soil was characterized as "mineral" while SOC (or M98) as "100% SOC". Is this correct?

*Indeed, SOC refers to simulation where soil is assigned as fully soil organic matter, and M98 turbulent flux option is used. Thus, SOC and M98 simulations are the same. As requested, we are going to change 'SOC' to M98 in the revised manuscript.*

L456 I don't think Figure 9 shows the ISBA-VS model version heavily overestimates accumulation, except 2021 for N-FOR.

*We will revise this by saying that the overestimation only occurs during 2021. This highlights the obscure results by ISBA-VS.*

L456-458 Can you provide a figure comparing two runs with between Wsw = 5 and 0.2? What does it physically mean? In section 2.3.1, the author mentioned "The coefficient Wsw relates to vegetation characteristics" but I don't think it would be enough to physically understand the parameterization and the results here. Please provide more details.

*Figure comparing Wsw = 5 and 0.2 is already provided in Appendix D, and is already at L459-L460: "The effect of snow cover fraction parameter wsw (see Sect. 2.3.1) for ISBA-VS snow depth simulations is detailed in Appendix D (Fig. D1)."*

*The idea of the Wsw parameter comes from the fact that the link between snow cover fraction and the vegetation characteristics are expected to be scale-dependent. Lower values of the parameter are expected at higher horizontal resolutions (LSM grid size). However as this parameterization has never been calibrated or evaluated against observed snow cover fractions, it can be seen from Table*

*D1 that this scale-dependence was not consistently applied among all applications, and that this parameter is sometimes adjusted on other criteria. This information was provided by developers of ISBA who have published the parameterization but without this level of detail, which may partly explain the lack of consistency among the different applications.*

*Reviewer 2 used "tweaking parameter" to qualify Wsw which is unfortunately common in numerical models. Our idea here is to emphasize these inconsistencies and to illustrate their impact on simulation results, see also our response to Reviewer 2. The discussion on that topic will be improved in the revised manuscript.*

L462-463 How did the authors know that the results are due to overestimated compaction and uncertainties in snowmelt? For example, have you compared snow density (as a proxy of compaction) with observations? If so, why did the errors of compaction and melt processes occur in terms of model physics - particularly snow energy processes?

*We also compared the simulated and observed snow water equivalent (SWE) on N-FOR but did not include the comparison in the manuscript as the SWE observations were not continuous (and not to further lengthen the manuscript). This SWE comparison will be included in the Supplement of the revised manuscript (see also Fig. 1 below), and it shows that the SWE is rather well simulated during the accumulation period. However, as the SWE does not fully explain the errors in HS simulations (underestimated peak HS and too early decrease in peak HS), the errors in snow simulations must also come from compaction. Uncertainties in compaction routine of Crocus are known as compaction due to metamorphism is not represented (Vionnet et al. 2012, Lafaysse et al 2017). Further, due to the interaction between compaction and surface energy budget, both errors may be linked. First, incorrectly simulated surface energy balance influences the compaction (e.g. warmer snow or higher liquid water content increases compaction). Secondly, errors in densification of snow have an impact on the surface energy budget as thermal conductivity of snow depends on density.*

[Figure]

*Figure 1. MEB simulations against observed height of snow (HS) and snow water equivalent (SWE) on N-FOR site.*

L464 Again, please include your relevant result in Supporting Info instead of "not shown". I think the authors would be able to discuss more in this section (3.3.2) because there are lots of interesting patterns such as the cold biases in fall. Can you add air temperature in this figure for comparison purposes? I'm guessing that the modeled soil temp seems to be much more sensitive to air temperature. Also, I would suggest adding discussion about the differences in soil temperature between ISBA-FS and MEB.

*Please see our earlier answer concerning all 'not shown' notions. We agree that air temperature could be interesting addition to the plot, but it would also require significant increase of the scale, making the soil temperature dynamics less visible. We are going to consider this for the revised manuscript. We are going to add the suggested discussion in the revised manuscript,*

L489-491 So how would the roughness length and turbulent exchange coefficient (CH) be used differently for soil and vegetation to minimize the overestimations of soil evaporation and sublimation?

*This is a good point and part of the motivation for developing MEB, so that the vegetation and soil can have separate roughness lengths, turbulent exchange coefficients, and thus, different magnitudes of turbulent fluxes. So, the answer is 'by using an explicit vegetation model' (see more details in Boone et al. 2017)*

Figure 12, for N-FOR, were the LSA observations missing or larger than 0.7? Please justify. To me, it is questionable why the LSAs from models were constant throughout the season, likely regardless of the existence of snowpack, which don't seem to make sense. Please discuss the potential causes.

*The gap in the N-FOR LSA observations is due to the polar night. The simulated LSAs for the forest are dominated by the albedo of vegetation that was assigned as constant (see Table 2.). The influence of evolving albedo of snow is thus not visible for the LSA.*

L498 "neglect" -> "do not estimate"?

*We will modify this in the revised manuscript*

L547-549 I don't think this statement is sufficiently supported by this study because the snow depth patterns from the models highly depend on year-by-year (Figures 5 & 9). "Wind erosion is higher in peatlands" is generally correct, but have you seen the wind data in the sites to see any wind effect?

*Thanks for pointing this out. We agree that the extent of wind erosion cannot be derived from the data, and the argument was too speculative. We will revise the text.*

Table 3 "Weakly stable" sounds somewhat weird. Please consider an alternative.

*The terminology used here is common for characterizing the stability regimes for atmospheric boundary layer flows (see e.g. Grachev et al. (2005, Boundary Layer Meteorology 116, p.201–235). We recognize that this may differ among research disciplines, however.*

L584-587 Does Crocus itself provide soil temperature simulation? I believe it only estimates snow and is coupled with ISBA to estimate soil parts. Thus it would be better to state something like "ISBA coupled with Crocus"

*You are right, the soil heat budget is solved in ISBA. We will revise this sentence to avoid confusion.*

L597 I don't think the authors explicitly showed soil evaporation from the results. Please rephrase it.

*Thanks for pointing out this inconsistency. Although we did not explicitly show soil evaporation in the results, the finding that the N-FOR latent heat simulations were improved when the vegetation fraction was increased (0.95 to 1.0, i.e all ET becomes controlled by stomatal), suggests that the soil evaporation plays a too big role. The sentence will be revised accordingly.*

*"We found ISBA-VS to drastically overestimate the LE, likely because of too high soil evaporation simulations due to its conceptualization of vegetation and snow cover fraction, resulting in too high LE."*

L717-718 Please reword

*We will modify this in the revised manuscript as:*

*"Our evaluation with the ESCROC ensemble Crocus snowpack model framework(ESCROC) gives confidence ensures that uncertainties in snow processes (not evaluated in this study) do not affect the robustness of our main conclusions summarized below."*

[Figure]

*Figure 2. Study site pictures (Lompolojänkkä (Pertti Ala-Aho), Siikaneva (Alekseychik et al. 2022), Kenttärova (Bastian Steinhoff-Knopp), Hyytiälä (Kolari et al. 2022)).*

[Figure]

*Figure 3. A) Study area locations inside the boreal land biome (green area, Olson et al., 2001), B) study sites locations in Finland (Source: Esri) and C-F) aerial images of each site (NLSF, 2020).*

References:

Grachev, Andrey A., Christopher W. Fairall, P. Ola G. Persson, Edgar L. Andreas, and Peter S. Guest. "Stable boundary-layer scaling regimes: The SHEBA data." Boundary-Layer Meteorology 116 (2005): 201-235.

Boone, A., Samuelsson, P., Gollvik, S., Napoly, A., Jarlan, L., Brun, E., & Decharme, B. (2017). The interactions between soil-biosphere-atmosphere land surface model with a multi-energy balance (ISBA-MEB) option in SURFEXv8-Part 1: Model description. Geoscientific Model Development, 10(2), 843–872. https://doi.org/10.5194/gmd-10-843-2017

Väliranta, Minna; Mathijssen, Paul J H (2021): Geochemistry of Siikaneva peat core from Finland. PANGAEA, https://doi.org/10.1594/PANGAEA.927689

Muhic, F., Ala-Aho, P., Noor, K., Welker, J. M., Klöve, B., & Marttila, H. (2023). Flushing or mixing? Stable water isotopes reveal differences in arctic forest and peatland soil water seasonality. Hydrological Processes, 37( 1), e14811. https://doi.org/10.1002/hyp.14811

Noilhan, J., & Mahfouf, J. F. (1996). The ISBA land surface parameterisation scheme. Global and Planetary Change, 13(1–4), 145–159. https://doi.org/10.1016/0921-8181(95)00043-7

Decharme, B., Boone, A., Delire, C., & Noilhan, J. (2011). Local evaluation of the Interaction between Soil Biosphere Atmosphere soil multilayer diffusion scheme using four pedotransfer functions. Journal of Geophysical Research Atmospheres, 116(20), 1–29. https://doi.org/10.1029/2011JD016002

Peters-Lidard, C. D., Blackburn, E., Liang, X., & Wood, E. F. (1998). The effect of soil thermal conductivity parameterization on surface energy fluxes and temperatures. Journal of the Atmospheric Sciences, 55(7), 1209–1224. https://doi.org/10.1175/1520-0469(1998)055<1209:TEOSTC>2.0.CO;2

Olson, D. M., Dinerstein, E., Wikramanayake, E. D., Burgess, N. D., Powell, G. V. N., Underwood, E. C., D'Amico, J. A., Itoua, I., Strand, H. E., Morrison, J. C., Loucks, C. J., Allnutt, T. F., Ricketts, T. H., Kura, Y., Lamoreux, J. F., Wettengel, W. W., Hedao, P., & Kassem, K. R. (2001). Terrestrial ecoregions of the world: A new map of life on Earth. BioScience, 51(11), 933–938. https://doi.org/10.1641/0006-3568(2001)051[0933:TEOTWA]2.0.CO;2

NLSF. (2020). National Land Survey of Finland Topographic Database. Available at: Http://Www.Maanmittauslaitos.Fi/En/e-Services/Open-Data-File-Download-Service.

---

## Author Comment (AC2)

The manuscript uses one new dataset over forests and peatlands situated in Northern and Southern Finland to evaluate different model configurations against surface energy fluxes, albedo, snow depth and soil temperatures. The authors emphasize the importance of testing these variables over organic layers and suggest which model configurations should or should not be used for some applications.

In its current form, not only is the manuscript too long, but it manages to be far too dense as well as too vague. There is not only too much information (e.g. there may "only" be 13 Figures in the main text and 3 in the appendices, but these 16 figures in fact include 86 windows altogether!), but also not enough detail where detail is needed. For example, many results seem to be relying on poorly understand parameters that are adjusted one way or another with little justification. I would not support the publication of this manuscript as it is, but I would strongly support it being split between a data paper and a modelling paper. This would not be "salami slicing"; the authors' express their intention for the data to be used in future modelling exercises and by other modelling groups. As such, the data can stand on their own. So can the modelling study which, despite needing more work and clarifications regarding the adjustments mentioned above, has the potential to convey important messages to the growing community of SURFEX (and other LSM) users on how not to misuse models.

*We thank the Anonymous reviewer for the detailed and critical opinion on our study. We are glad that the Reviewer appreciates our model evaluation and sees this study as a potentially significant contribution to the LSM community. We also recognize the comment that the dataset itself has value for other modelling studies. Instead of a data paper, we have decided to publish the datasets and metadata to open access data repository.*

*As requested, we will attempt to reduce the number of figures/panels in the main manuscript. However, please note the comment of Reviewer 1 who specifically appreciates the completeness of the results and points out the high quality and information content of the Figures. Thus, we prefer to move some panels into Supplement rather than fully removing them. In respect to both reviewers, we have considered alternative ways to shorten the manuscript without deteriorating the quality of the content.*

*Alternative (1)*
*Our first and preferred alternative to shorten the main manuscript is as following:*

- *As the LE flux was shown to have a minimal role in Fig. 4 we are going to remove the panels showing LE in Fig. 6. The current Fig. 6. will be provided in the Supplement.*
- *As the sensitivity of snow depth to SOC is not major, we are going to remove the height of snow panel in Fig. 8. However, in reply to a request from Reviewer 1, we will provide plots with HS sensitivity to SOC (both sites) in the Supplement. In Fig. 8 we are also going to reduce the presented soil depths to include only 5cm, 25cm and 70cm. Plots with other soil depths will be moved to the Supplement.*
- *The impact of the soil-vegetation representation can be demonstrated with fewer soil depth panels. Thus in Fig. 10, we are going to keep only N-FOR 5cm and S-FOR 5cm soil depths in the main manuscript. The current figure will be moved to the Supplement.*

*This way we are going to reduce the number of panels in the main manuscript by 12. In addition, we will attempt to shorten the text when preparing the revised manuscript.*

*(Alternative 2)*
*The second alternative that we have considered is to move all scatterplots to the Appendices and to present these results as tables of performance metrics in the main manuscript. However, we believe that model behaviour is more difficult to understand from the metrics alone, and that this change would have a slightly deteriorating impact on the overall quality of the paper. For instance, the metrics alone do neither represent the scale nor the range of the fluxes.*

*Therefore, we kindly ask the editor to choose between these two alternatives to best respect both the reviews.*

*We will also even more critically point out the 'free' parameters / poorly represented processes in the evaluated model combinations. We agree with the reviewer that the changes suggested below will improve the manuscript and increase its value for the SURFEX and LSM modelling communities.*

*We would like to point out and apologize for two mistakes found in the data processing during an additional quality check motivated by the reviews. Thus, we would like to thank the reviewers for boosting our data quality control process.*

*First, we found 2h discrepancy in time zones between different data sources, notably between the main observations used for forcing and the data used for gap-filling. We have converted all the forcing and evaluation data into UTC time zone and ran new simulations. We will update the figures to include the corrected simulations in the revised manuscript. These changes do not have any major impact on the results of the study.*

*The second error was in the wind forcing reference height used for the S-WET site. The S-WET wind reference height had been assigned as in the contiguous S-FOR site (16.8 m), whereas in reality wind was measured at 3 m height. This means that simulated wind speed near the surface of the low-vegetated S-WET site was unrealistically low in our simulations. We have found that this reduces the sensitivity of turbulent fluxes (mainly sensible heat exchange) simulations to different turbulent exchange parameterizations, as the site experiences stable conditions less often than previously simulated. This correction does not change the results showing that M98 turbulence option is superior to classic stability correction function (RIL). However, this correction does change the snow depth simulations on S-WET. Figure 5. changes in a way that all the turbulent exchange parameterizations succeed in simulating the snow melt (see the revised Fig. 5 below). Results related to these changes will be corrected in the revised manuscript.*

[Figure]

*Revised Figure 5. Time series of snow depths simulated by ISBA-ESCROC. The 35 ensemble members are grouped by their turbulent flux parameterization, and the spread of each group is presented in colored ranges. Observed snow depths are presented in black dots and dashed lines.*

**The major comments are separated into data and model:**

Data:

- L311-312. How similar/different are the contiguous sites? And the meteorological stations? How much of the radiation data were missing? By R_g and R_A? How many timesteps were filled by ERA5? Ideally, either the timeseries of all meteorological observations or scatterplots showing how much these different sources differ when we do have overlapping data should be included. As this manuscript promotes a brand new dataset used for model evaluation, the gap filling in the dataset cannot be brushed off in two sentences.

*We agree that there was not enough detail on the dataset. We are going to publish the datasets used for model forcing and evaluation (see also our answers to your comments about data publication later).*

*In the published model forcing metadata, we are going to add summary of how much each site variable was filled with different datasources. Citation to this dataset will be added to the revised manuscript.*

*The contiguous sites refer to the sites used in this study (e.g. contiguous site of N-FOR is N-WET and vice versa). Although they are different ecosystems, locations are close to each other and meteorological conditions are similar. The "other nearby meteorological station" used in gap-filling are farther; Sodankylä (ID 101932) is ~120 km from N-WET/N-FOR and Ähtäri (101520) is ~80 km from S-WET/S-FOR.*

*In case of N-WET, N-FOR and S-WET, ERA5 data in forcing is very minimal (less than 10 hours). S-FOR uses more ERA5 data due to lacking and discontinuous radiation observations. Specifically, the*

*downward longwave radiation observations were lacking in 2008 to 2010 and again in 2012. As S-FOR forcing data includes more ERA5 data than the other sites, we are going to include a comparison of ERA5 against site observations on S-FOR in the Appendice of the revised manuscript. Note that we did not focus on those years with ERA5 data in the manuscript, but they are still included in the scatterplots and performance metrics.*

*We are going to revise lines L310-L311 as:*
*"The data gaps in meteorological observations were first filled by the contiguous sites (e.g. contiguous site of N-FOR is N-WET and vice versa) and the remaining gaps by other nearby meteorological stations (IDs: N-WET/N-FOR 101932, S-WET/S-FOR 101520)."*

- What area does the footprint of the eddy covariance towers cover? Does the vegetation cover or topography vary within the footprint? May this have consequences on the measurements? These questions may have been answered in the two cited papers, but such information is needed here.

*We agree it is imperative that eddy-covariance (EC) fluxes used for model evaluation are both representative of the system measured and meet the fundamental micrometeorological assumptions (i.e. turbulence stationarity and horizontal homogeneity of the source-sink function). In the data used here, these are ensured by the original authors of the datasets, who have conducted both standard Q/A routines and footprint analysis. For respective sites, these details have been reported in earlier publications and data descriptions (Aurela et al. 2015, Mammarella et al. 2016, Mammarella et al. 2019, Alekseychik et al. 2022). As the data quality is ensured, and follows the standards of the FluxNet community, we believe that as data users, we should be able to trust the data and rely on the uncertainty estimates provided by the data authors.*

*Considering that the main goal of our paper is a detailed model evaluation, we have decided to not include flux footprints and/or deeper insights of the data processing etc. in the current paper. The relevant information is available in the cited publications, and in the current manuscript we focus on the modelling part. In the revised manuscript we will further improve this part by pointing out the key papers and dataset descriptions relevant. Note also that uncertainties associated with EC measurements were considered and discussed in our manuscript based on the content of these papers.*

- Much of the appendices could be transferred to a paper describing the data. Same for Section 4.1.
- L665-676. This could be added to a data paper.

*As noted above, we prefer not to split the paper.*

- L735-737. The dataset presented here will not be widely re-used unless it is published in a data repository and a separate manuscript details the data. In a research landscape where open access to datasets and models is required by many funders and is often a pre-requisite for publication (I am surprised it is not compulsory in TC), a sentence like "Data are available upon request from the authors" raises red flags. You have done all the work on the dataset; with FAIR guiding principles becoming the norm rather than the exception, not publishing it suggests that the data are perhaps not as solid as should be.

*We thank the reviewer for this comment. As the sites we consider in this study belong to several international and national research infrastructures (e.g. ICOS; S-FOR, S-WET, N-FOR, N-WET), the*

*data used here is mostly already citable and openly available in databases. For instance, the data for S-FOR and S-WET can be accessed through https://smear.avaa.csc.fi/download under fair-use principles. The ownership of the data is by the original data provider organizations and groups. However, we fully agree with the reviewer that sharing the particular dataset used in this paper (including measured fluxes, state variables, site properties and model forcings) is likely to catalyst further use of the data in land-surface and hydrological model development, by saving time and effort of other modellers – and reducing potential mistakes in data processing/filtering/use. This is not least because compiling the dataset required specific insights into the observations, sites and vivid communication with the researchers responsible for the measurements. This is clearly the strength of the author team in this study. Therefore, we got permission to publish the dataset specific to this study, accompanied by relevant metadata descriptions, in open access data repository Fairdata.fi as an electronic supplement to this article with an assigned DOI for the dataset.*

Model

- As acknowledged by the authors, not only is w_sw important, but it is very poorly defined and, arguably, poorly understood. Table D1 clearly shows that w_sw is what I would call a "tweaking parameter"; it can range from 0.1 to 5 for no particular reason it seems. In addition, there does not seem to be any explanation given by the authors as to why different w_sw make so much difference in N-FOR but not in S-FOR. This needs to be explained, and preferably not in an Appendix (in fact, I am unclear as to what criteria were used to choose appendices over core text).

*We agree with the reviewer that the parameter w_sw is, indeed, a tweaking parameter with very limited physical interpretation and seems not to be well related to any measurements. However, existing literature has never emphasized its high sensitivity on simulation results and its inconsistent settings among various applications. This is exactly what we pointed out in this paper and tried our best to explore the sensitivity of model results to the seemingly arbitrary choice of w_sw in the earlier studies. Please see also our answer to Reviewer 1.*

*The snow cover fraction, influenced by the w_sw parameter, affects the turbulent fluxes and the ground-snow heat flux. This approach commonly reproduces unrealistic snowpacks and soil thermal regimes in forested areas (Napoly et al. 2020) as the turbulence of the rough vegetation directly alters the soil-vegetation temperature which then have a cooling or warming impact on the snowpack. Although the w_sv parameter do make a difference on both sites, the sensitivity of w_sv parameter for the soil thermal regime is higher on N-FOR (see Fig 1. below). During some warm events on N-FOR, the higher snow cover fraction simulation (w_sv = 0.2) retains freezing soil temperatures while the soil of the lower snow cover fraction (w_sv = 5) simulation is no longer freezing, melting the snowpack. In the revised manuscript, we will discuss this differing feedback.*

[Figure]

*Figure 1. The sensitivity of w_sw parameter on 6-hour surface soil temperature simulations and snow water equivalent (SWE). Simulations are represented by one member of the ESCROC-ISBA VS as described in the paper.*

*We agree that many of the supporting results, now placed in Appendices, could have been placed in the main text. However, our study contains already a lot of figures, as the Reviewer has also pointed out. Our approach was to include supporting results as Appendices. For instance, Fig. B1 supports our discussion of snow in forests versus snow on peatlands while Fig. C1 is supporting our discussion about uncertainties of winter vs. summer energy fluxes (we obviously focused on winter in the main manuscript). When it comes to the results of sensitivity w_sw parameter sensitivity test, we believed it to be very useful information for the expert model users, but not essential for the main manuscript.*

- One of the premises of this manuscript, with which I agree, is that peat and SOC are generally overlooked (l78-89) in snow modelling studies. L90: "The goal of this study is to evaluate the ability of SURFEX LSM (Surface Externalisée, Masson et al., 2013) to describe the surface energy balance and its drivers in boreal and subarctic peatlands and forests". Yet, l290, we learn that the authors will not reach their goal by using a dataset fit for purpose, but instead "assumed the top 1 m of the peatlands to consist 100 % of SOC". How can your whole study rely on an assumption?

*Boreal and sub-arctic peatlands, as the fens in our study, have typically a rather deep (down to several meters) peat layer overlying the mineral soil or bedrock (Marttila et al. 2021). The peat consists purely of organic plant residues decomposed to different degree. For instance, Väliranta and Mathjissen (2021) measured mean loss on ignition value 96,6 % in the first meter of the peat profile at S-WET.  Further, Muhic et al. (2023) also measured organic matter to be nearly 100% for the first 40 cm depth at a location just next to the N-WET peatland (see Muhic et al. 2023 Fig. 1b "FP2").*

*Thus, in this respect our assumption of '100% SOC' is realistic. We are going to revise the manuscript by adding these citations to justify the use of 100% organic matter in model parameterizations.*

*However, we recognize that particularly hydraulic characteristics are likely to vary across the peat profile and this variability is not detailed in the used model schemes. The thermal characteristics (heat capacity and conductivity) in the organic, saturated or near-saturated peat soils with porosity >0.8-0.9 are however dominantly determined by liquid water and ice contents.*

- Table 2 suggests a fixed vegetation for ISBA, but nothing for MEB. How does MEB know how much of the grid box is vegetation and how much is vegetation air space? Does it change over time? What are the implications? Please clarify.

*In MEB, the vegetation input includes vegetation type, LAI and canopy height. Vegetation properties used in the different canopy process parametrizations are derived from these inputs.*

*In the paper, L281-283: "Summer LAI and vegetation height were obtained from literature, while winter LAI was estimated according to the proportion of deciduous and coniferous vegetation on each site.". It is possible to provide monthly LAI and canopy height values to account for seasonal variability in foliage density. We have estimated the monthly LAI cycle between the LAImin and LAImax (Table 2.). Indeed, such estimation, together with the known challenges in in-situ LAI estimates brings uncertainties. We are going to publish the parameter-files (so called namelists) as electronic supplement of this article.*

*The 'vegetation fraction' parameter as in ISBA, does not exist in MEB. However, the part of the vegetation through which light passes without hitting the leaves is computed using a so-called sky view factor which depends on LAI and a vegetation dependant -constant (see Boone et al. 2017 Eq. 45). This means that the larger the LAI, the less light goes through (i.e. the vegetation air space is smaller).*

*We will revise the manuscript to be clearer in these regards.*

- In Section 4.4., the authors acknowledge that the lack of internal water vapour causes errors in heat exchanges between the snow and the soil and therefore potentially affects modelled soil temperatures. Knowing this, can the authors demonstrate that accounting for (an assumed) SOC is not a way to compensate for errors in the soil thermal regime that are caused by other processes that are badly or not represented?

*The implementation of organic content in the solving of thermal diffusion was done considering the known thermal properties of this material, without any specific calibration of parameters to reproduce soil temperature observations. This way, we can reasonably assume that the improvement obtained in soil temperature simulations is obtained for an appropriate physical reason. Obviously, errors and error compensations due to unresolved processes always remain in any numerical model and it is difficult to demonstrate their influence or absence of influence without implementing these unsolved processes, (and vapor transfer is an unsolved process in most state-of-the-art LSM). This intrinsic limitation of any numerical model will be mentioned in the revised manuscript. See also our response to reviewer 1.*

- There are far too many plots that are not even referenced in the text. Then there are performance metrics in the plots. The more is not the better. Please reconsider whether you need 86 windows/plots in 16 figures and consider presenting your results in line with what you are highlighting in the text. For example, the different parts of the energy budget are important in different seasons, so why not have seasonal plots? You are asking the reader to

compare seasonal plots (l516-517), it is your role to facilitate this if you believe this is important.

*We thank the reviewer for pointing this out. Please see our answer earlier regarding our effort to reduce the information content of some figures. However, again, please note the comment of Reviewer 1 appreciates the completeness of the results and the high quality of the figures. Therefore, we also prefer to move some Figures in the Appendix and/or Supplement rather than removing information that we believe is necessary to fully understand the model behaviour for the most expert readers. Further, we believe the performance metrics and plots support each other. The performance metrics provide quantities to otherwise qualitative illustrations of the figures, important in understanding the model performance. See also the answer below regarding Figure 7.*

*We are going to revise the manuscript at L516-517 as:*

*"The relative uncertainties in simulated and observed energy fluxes are significantly greater in winter than in summer. Performance of the simulated summer energy fluxes is very good (see Appendix C Fig. C1). (compare Fig. 13 vs. Appendix C Fig. C1)."*

**Minor comments:**

The list of symbols and acronyms is huge and it makes it very hard to follow the manuscript, having to go back to previous pages to remember what is what. I would strongly advise the authors to make it easier for readers by having one or multiple tables describing the abbreviations, acronyms etc. It may also help the authors catch some that are not described (e.g. rho_sng).

*We are going to add a Table describing the abbreviations and acronyms.*

Abstract: I disagree that the model included a "realistic" soil texture; the SOC values were assumed, not "real". Please re-phrase.

*Considering these peatland soils as 100% organic matter is indeed realistic as we have explained in the 'Model' comments.*

L85: Incorrect reference. Krinner et al. (2018) do not provide any information at all about soil texture at ESM-SnowMIP sites; Menard et al. (2019) do.

*Thank you, this will be corrected.*

Section 3.3.4: The models neglect LSA increases due to intercepted snow, but does intercepted snow sublimates? I could not find the answer in the manuscript even though a large percentage of snowfall is known to sublimate in coniferous forests (see Essery and Pomeroy, 2001 http://www.merrittnet.org/Papers/Essery_Pomeroy_2001.pdf for references).

*This is a good point, and yes, intercepted snow can sublimate in MEB. This will be mentioned in the revised manuscript.*

Eq 9: Given how important snow density (rho) is to the calculation of the snow effective thermal conductivity, it would be helpful to know how rho is calculated.

*You are right, snow layer densities are the main driver of snow thermal conductivity. However, it is not possible to provide a comprehensive description of all snow processes in this paper focused on*

*model application and evaluation. To not make the manuscript longer, we are going to add detailed citations to the model papers and equations numbers describing falling snow density, compaction, and associated interacting processes.*

Figure 1: What is the point of the top plot? We can hardly see where the sites are located in Finland, which would be more interesting than knowing where the boreal land biome is in the whole world. Also, could you please indicate the scale of the aerial images? Do they cover the EC tower footprint? If the scale is larger than the footprint, then, again, what is the point? The images should be proportional to how the sites are used. The manuscript presents site simulations, therefore we should have an idea of what the sites looks like. Otherwise, scrap Fig 1 altogether and simply present the sites as the parameters that represent them in the model.

*The goal of the top plot is to contextualize our study sites inside the boreal land biome and to illustrate the extent of this biome around the Northern hemisphere. We have revised Fig. 1 to include site locations inside Finland and the aerial images to include scale bar (see Fig. 3 at the end of this document). In addition, we are going to add site photos to the Supplement (see Fig. 2 at the end of the document)*

Fig. 3: Add "Observed" at the start of the caption. Also, G is hardly visible

*Thanks for pointing out, will be corrected. G is hardly visible because its magnitude is very low compared to other fluxes. This will be highlighted in the revised manuscript.*

Fig 7 and others: Do we really need scatterplots, qqplots and timeseries? They are not all referenced in the text. If you want to use them all, please explain why, but I would advocate choosing.

*We believe that the use of time series plots and scatterplots support each other: time series qualitatively illustrate part of the simulations (when and what are the errors) while scatterplots gather all the data points for the full simulation period (whether or not time series patterns are consistent or relevant for the full simulation). Qqplots clearly identify the mean biases for the full obs-mod sorted distribution. We understand that it may take a while for reader to process all this information, but we do not think this would be a valid reason to remove some of them. Indeed, the impact on snow cover simulations can be very different between systematic errors and time-varying errors of energy fluxes.*

Fig 7: Why did you choose that specific year for the timeseries?

*This winter represents typical snow conditions for both sites, and good availability of upwards radiation observations. This will be mentioned in the revised manuscript.*

L442: Same as in the abstract. I disagree that the model included a "realistic" soil profile; the SOC values were assumed, not "real".

*See our answer in 'Model' comments.*

L493: "the summer energy fluxes were majorly improved by simply assigning the vegetation fraction to unity". Is there a legitimate reason to assign the vegetation fraction to 1, or is it a tweak to "improve" the energy fluxes albeit for the wrong reasons?

*You are right, this is a way to tweak the ISBA composite approach, and it was done by e.g. Vernay et al. 2022. Vegetation fraction is commonly applied as 0.95 in ISBA, which is by no means physical either. The benefit of an accurate vegetation fraction depends on the way interactions between*

*vegetated and non-vegetated parts are represented. However, assessing this was beyond the scope of this study.*

*Our point is that such composite approach is only very lightly linked to physical relationship between soil and vegetation. We presented alternative ways to optimally apply such model to these four environments.*

L566: Do you mean Menard et al (2021)?

*Yes, we apologize for this incorrect reference which will be corrected in the revised manuscript*

L584-592: This is a very important paragraph on how not to misuse or repurpose models. Splitting this manuscript into one data description and one model simulations paper would prevent such an important message from being buried deep under too much information. I would also like to see this message somewhere in the abstract.

*We thank the reviewer for the importance he/she has identified in this paragraph. We have explained earlier in this response why we prefer to not split the paper in two parts, but indeed we will insist on that point in the conclusions and abstract to better emphasize this message.*

Sections 4.3.1 + 4.3.2.: What I called "tweaking", the authors call "compromise". These sections are very honest about how some of the results were "improved" and are, in my opinion, the best in the manuscript. Would the authors consider be this transparent earlier in their manuscript?

*Thanks for the comment; we agree the importance of identifying and openly communicating the critical 'tweaked' / poorly constrained parameters in LSM's. Throughout the revised manuscript, we plan to be even more transparent and clear on such inadequately described processes, and parameters that cannot be readily derived from observations / available data. These include, for instance, the alternative formulations for turbulent exchange in stable conditions (Sect. 2.3.2, Figs. 4, 5, 6). We believe being more critical to the model schemes, as implicitly suggested by the reviewer, will further improve the manuscript and make it more useful contribution to SURFEX, snow and LSM model communities.*

Figure 12: This is very confusing. The legend convention is the opposite of Fig 4. where solid colours are the model, dashed are observations... Please be consistent.

*As requested, we are going to change the dashed lines to solid.*

L682-683: Does this mean that you may have achieved "satisfactory model performance" for the wrong reasons i.e. because one badly represented process compensates for another not represented at all (e.g. overestimating snow density may cause snow depth to be as low as if the model had accounted for lateral snow transport).

*With the available processes implemented in the model, it is again not possible to be affirmative on error compensations. Here, the parameterization of wind impact on snow density is designed to replace the fact that snow transport is not explicitly simulated. Regarding snow density, error compensation with falling snow density, and mechanical snow compaction are obviously possible but difficult to identify without more detailed observations. Regarding snow height, error compensation between snow density and snow mass are indeed also possible, with a part of snow mass errors that can be explained by the absence of erosion/accumulation in the model. Our goal in our discussion is*

*to help the reader to easily identify the unresolved processes we can imagine having significant impacts on our results, and to be aware of most possible error compensations.*

*Despite these limitations, we believe that it is also fair to present results as satisfactory when the magnitude of snow height and energy fluxes is realistic, even if not always for the good reasons, because realistic estimates of snow extent and energy fluxes is an important criteria to have realistic boundary conditions in NWP systems and GCM.*

L699-701. Information about the footprint of the EC tower should be given earlier in the manuscript.

*Please see our response to the general comments earlier.*

[Figure]

*Figure 2. Study site pictures (Lompolojänkkä (Pertti Ala-Aho), Siikaneva (Alekseychik et al. 2022), Kenttärova (Bastian Steinhoff-Knopp), Hyytiälä (Kolari et al. 2022)).*

[Figure]

*Figure 3. A) Study area locations inside the boreal land biome (green area, Olson et al., 2001), B) closer study sites locations in Finland (Source: Esri), and aerial images of each site: C) Lompolojänkkä (N-WET), D) Kenttärova (N-FOR), E) Siikaneva (S-WET) and F) Hyytiälä (S-FOR) (NLSF, 2020).*

References

Marttila, H., Lohila, A., Ala-Aho, P., Noor, K., Welker, J. M., Croghan, D., Mustonen, K., Meriö, L., Autio, A., Muhic, F., Bailey, H., Aurela, M., Vuorenmaa, J., Penttilä, T., Hyöky, V., Klein, E., Kuzmin, A., Korpelainen, P., Kumpula, T., … Kløve, B. (2021). Subarctic catchment water storage and carbon cycling – Leading the way for future studies using integrated datasets at Pallas, Finland. Hydrological Processes, 35(9), 1–19. https://doi.org/10.1002/hyp.14350

*Väliranta, Minna; Mathijssen, Paul J H (2021): Geochemistry of Siikaneva peat core from Finland. PANGAEA, https://doi.org/10.1594/PANGAEA.927689*

Aurela, M., Lohila, A., Tuovinen, J. P., Hatakka, J., Penttilä, T., & Laurila, T. (2015). Carbon dioxide and energy flux measurements in four northern-boreal ecosystems at Pallas. Boreal Environment Research, 20(4), 455–473.

Mammarella, I., Peltola, O., Nordbo, A., & Järvi, L. (2016). Quantifying the uncertainty of eddy covariance fluxes due to the use of different software packages and combinations of processing steps in two contrasting ecosystems. Atmospheric Measurement Techniques, 9(10), 4915–4933. https://doi.org/10.5194/amt-9-4915-2016

Mammarella, I., Rannik, Ü., Launiainen, S., Alekseychik, P., Peltola, O., Keronen, P., Kolari, P., Laakso, H., Matilainen, T., Salminen, T., & others. (2019). SMEAR II Hyytiälä forest eddy covariance.

Alekseychik, P., Peltola, O., Li, X., Aurela, M., Hatakka, J., Pihlatie, M., Rinne, J., Haapanala, S., Laakso, H., Taipale, R., & others. (2022). SMEAR II Siikaneva 1 wetland eddy covariance.

Muhic, F., Ala-Aho, P., Noor, K., Welker, J. M., Klöve, B., & Marttila, H. (2023). Flushing or mixing? Stable water isotopes reveal differences in arctic forest and peatland soil water seasonality. Hydrological Processes, 37( 1), e14811. https://doi.org/10.1002/hyp.14811

Vernay, M., Lafaysse, M., Monteiro, D., Hagenmuller, P., Nheili, R., Samacoïts, R., Verfaillie, D., & Morin, S. (2022). The S2M meteorological and snow cover reanalysis over the French mountainous areas: description and evaluation (1958-2021). Earth System Science Data, 14(4), 1707–1733. https://doi.org/10.5194/essd-14-1707-2022

Napoly, A., Boone, A., & Welfringer, T. (2020). ISBA-MEB (SURFEX v8.1): model snow evaluation for local-scale forest sites. Geoscientific Model Development, 13(12), 6523–6545.

Olson, D. M., Dinerstein, E., Wikramanayake, E. D., Burgess, N. D., Powell, G. V. N., Underwood, E. C., D'Amico, J. A., Itoua, I., Strand, H. E., Morrison, J. C., Loucks, C. J., Allnutt, T. F., Ricketts, T. H., Kura, Y., Lamoreux, J. F., Wettengel, W. W., Hedao, P., & Kassem, K. R. (2001). Terrestrial ecoregions of the world: A new map of life on Earth. BioScience, 51(11), 933–938. https://doi.org/10.1641/0006-3568(2001)051[0933:TEOTWA]2.0.CO;2

NLSF. (2020). National Land Survey of Finland Topographic Database. Available at: Http://Www.Maanmittauslaitos.Fi/En/e-Services/Open-Data-File-Download-Service.

---

## Author Response (AR2)

Thank you for the constructive review. We believe the manuscript was greatly improved following these suggestions. Please find all our answers below in blue.

Thanks for your amendments to the manuscript. The inclusion of data not shown as supplementary material and removal of some subplots is hugely beneficial. Although your response to reviewers indicated you would revise the text to make this shorter (and I requested a refocus on the main messages of the paper), it does not appear to be shorter. The discussion currently covers almost 7 double-spaced pages and this does the work presented here a disservice. Please could you revisit the text here. Some of the discussion repeats methodology and results (e.g. lines 535-539, 548-551,..613..) and this can be cut. Please aim for around 3-4 pages for the discussion.

Thank you. We have now revisited the discussion, and removed the repetition and restructured the paragraphs around the main messages. Now the reader can find the main message as a starting sentence for each paragraph. That is followed by discussion related to existing literature and justification for arguments. The revised discussion makes up roughly 4 pages.

Please also think about the methodology and how this could be reduced to focus the readers attention - for example lines 214-218 including equation 9 isn't needed. As ISBA, MEB and CROCUS are reasonably well-known, some details can be left to the reference. However, turbulent transfer, vegetation differences and soil heat transfer details are important as these form part of the (focused) discussion so need to stay.

We have reduced the methodology as requested. We agree that the Eq. 9 is not absolutely necessary in the paper and have removed it. We also identified other instances where to make the section more compact: combination of some equations (Eq. 1+2 and Eq. 5+6), and removal of equations were conducted (Eq. 11, 14). Also, some parts of the text were shortened.

Although this will lengthen the manuscript slightly, please remove the acronyms LSA, MOST, HS, NWP and GCM. This is a worthwhile trade to aid readability. Define what you mean by albedo early and use albedo thereafter - you can use the discussion to refresh what you mean by albedo. MOST is only used twice I think. Snow depth is easier to read than HS. It's fine to use these acronyms in the figures - as long as you define them in the captions. Also move definition of LSM out of the abstract.

We understand that removing acronyms can improve the readability. Thus, we have revised the paper accordingly:

- MOST removed and now written always as the Monin-Obukhov similarity theory
- Land surface albedo defined now as albedo. LSA acronym is kept in equations and figures. Land surface albedo is mentioned again in the discussion to refresh the reader.
- HS removed from the text but this abbreviation is maintained in the figures and equations as this is the official vocabulary and notation from the International Classification for seasonal snow on the ground (Fierz et al., 2009 ; https://unesdoc.unesco.org/ark:/48223/pf0000186462).
- Acronyms NWP and GCM removed altogether.
- LSM removed from abstract.

A shorter, more succinct manuscript will be more impactful. There's certainly far more to pick out than can be presented - it's worth sacrificing some content to broaden the audience. Those who care about the details will contact you! Some specific comments are given below.

*We appreciate your feedback and have done our best to deliver an improved manuscript in these regards.*

Other specific recommendations are:

line 8 'We tested the sensitivity...': make this more explicit e.g. tested different turbulent flux representations. Although this includes the ESCROC subset 2 parameterisations, this is not a focus of this paper. Section 2.3.2 first paragraph is sufficient.

*The sentence revised as:*
*"We tested different turbulent flux parameterizations…"*

line 39-40 move the definition of energy flux acronyms into the methodology

*Introduction and methodology were revised as requested. We revised the presentation of model acronyms with same logic.*

line 99-101 'On the ... sites' -> 'At the ....sites'

*These prepositions were revised as requested.*

Section 2.1.1. Title: add '(N-WET and N-FOR)'

Section 2.1.2. Title: add '(S-WET and S-FOR)'

*Titles were edited accordingly.*

line 127 'N-FOR' -> 'S-FOR' also 'is managed' -> 'is a managed'

*Thank you for pointing out the wrong site code. However, it seems that it was already written as 'is a managed'*

line 136. Bring in Figure 2 at this point and state ISBA-FS with Crocus is used at all sites, ISBA-VS and MEB only for forest sites.

*We revised this part by being clearer about the use of ISBA coupled to Crocus and MEB coupled to Crocus at different sites, with references to Fig. 2:*
*"Specifically, ISBA coupled to Crocus is used for both peatland and forest site experiments (Fig. 2A,B) whereas MEB coupled to Crocus is only used for the forest site experiments (Fig. 2C).}"*

*However, we did not use ISBA-FS and ISBA-VS definitions but instead referred to model configurations that are defined later on:*
*"Parameterizations and different configurations of ISBA, MEB and Crocus models are detailed later in Sect. 2.3"*

line 155. Use this opportunity to reinforce C_H being critical by adding 'and is one of the parameters that is the focus of this study' or similar to the end of the sentence.

*Good point, this was revised.*

line 237. Remove 'so-called'

*This was removed.*

line 240 add 'p_sn' after effective snow cover fraction - although it equation 11 needed? It's useful to know the proportion of snow over vegetated or bare soil is governed by snow depth (can be said explicitly in text after 'atmosphere' in line 239)

Equation 11 was removed as the computation of effective snow cover fraction is already defined in the text as "The effective snow cover fraction is the weighted average between the snow fraction of vegetation (p_snv) and snow fraction of the bare ground (p_sng)". We now mention that the fraction is governed by snow depth.

line 241 clarify 'bare ground' rather than 'ground'

This clarification was added.

Figure 2 consider switching the order from A, B, C to C, B, A i.e start with general and simplest case, then increase in complexity for the forest.

Figure order was switched as requested. It indeed makes more sense in this order. We have also edited the order of presenting these cases in the text.

Section 2.3.2. Define ESCROC-E2 for the E2 subensemble and use thereafter in place of ESCROC? Line 285-287 can then be simplified (and remove 'while')

ESCROC-E2 subensemble was defined and used thereafter.

line 295-296 remove 'BONE' and 'BOGR' acronyms. It's fine to leave them in Table 2 as they have already been defined in the caption.

These acronyms were removed from the text.

line 354-358. Shorten: both cases use site parameters and default ESCROC-E2 but differ in treatment of turbulent transfer and soil representation.

This was shortened as requested.

line 421. N-FOR isn't shown in figure 5. Is this a typo?

Thank you pointing this out. It was a typo that is now corrected to N-WET.

Figure 6 caption should be LWU rather than LE?

You are right, this was another typo that is now corrected.

line 500 'LSA is underestimated....' where is this shown? It is not in Figure 11. If this relates to section 3.3.4, the text from 'LSA is...' to 'Sect. 3.3.4' can be removed.

It is shown in Fig. 11 A): SWU of the lower end (0 to 75 Wm$^{-2}$) is underestimated by the model. This is also visible as underestimated albedo during winter in Fig. 12.

line 652-656 belongs in the introduction (if anywhere)

This was removed completely.

line 763 'in expense' -> 'at the expense'

This was corrected.

---

## Author Response (AR3)

Thank you for the review and acceptance. We have made the requested corrections. Please find our answers below.

Many thanks for the revisions to your paper. This is now much more succint, with clear implications for the numerical weather and hydrological prediction communities. This paper should be published subject to the following technical corrections:

Line 193. Remove 'so-called' and change 'vegetation dependant -constant' to 'vegetation-dependent constant'

Thank you for pointing these out. They are now corrected as requested.

Line 341. Change 'In total of ca. 290....' to 'In total, 290...' or 'A total of 290...'. Is there some uncertainty around the number of simulations that were conducted? I just wonder about the use of ca. It's better to be precise if you can.

Indeed the 'ca 290' was too vague and actually did not include simulations that were only plotted for the supplement (35+35 simulations of ISBA-VS with wsw = 0.2, and a simulation where vegetation fraction was assigned to unity). We also noticed that these simulations were not mentioned in the Section 2.5. (Model Experiments). These are now added at lines 337-342 (see small changes for this paragraph):

*"On the forest sites, we examine the skills of the different alternatives to represent the energy and mass budgets of soil and vegetation (ISBA-VS, ISBA-FS, MEB in Sect. 2.3.1), and their implications on snow depth, soil temperature and surface energy fluxes. First, we compare ESCROC-E2 simulations with these three configurations focusing on the snow depth and soil temperature. These ISBA-VS simulations are conducted with the default snow cover fraction parameterization (wsw = 5 in Eq. 8). Additional ISBA-VS simulations are performed to assess the sensitivity of the wsw parameter, using a value of 0.2 in Eq. 8. For a more detailed comparison of the simulated and observed above-canopy surface energy fluxes by ISBA-VS and MEB, we conducted deterministic simulations with both ISBA-VS and MEB, the default snow cover fraction and Crocus parameterizations (as in Fig 2. in Lafaysse et al. (2017)). Additionally, we perform a deterministic ISBA-VS simulation with the default snow cover fraction parameterization, but the vegetation fraction set to unity."*

Then, we have also corrected the simulation count to be more precise and to contain all the simulations that were necessary to produce the paper (including supplement):
*"A total of 361 ensemble and deterministic simulations were conducted."*

Please run all figures through a colourblind checker and amend colours as needed. Figure 5 and 9 are hard to decipher under Blue-blind / Blue-weak configurations (https://www.color-blindness.com/coblis-color-blindness-simulator/). I did not check them all.

Thank you for the colour check. We have now checked all the figures with following changes:

- The red colour of Figs 9, 10 and S3 was darkened.
- Colours of Fig. 5 were updated. This one remains a bit challenging as it has multiple overlapping colours but the updated colour combination is clearly better.

Line 585. Please rephrase 'we found MEB to systematically simulate too early snowmelt' e.g. to 'we found MEB to simulate snowmelt too early' ('systematically' is tricky to place grammatically and I don't think it adds anything).

Thank you, we have corrected this as requested. In addition, we changed our writing of 'snow melt' to 'snowmelt' throughout the paper.

Line 622. 'miss-match' -> 'mismatch'

This was corrected.

Finally a comment on line 72 - no changes needed because this statement is technically true as far as I'm aware. However, the phrasing makes it suggest that it is not worth undertaking a literature search, whereas there is more out there. Although this paper doesn't focus on turbulent transfer or soil properties, if you or anyone reading this would like to access the dataset on subcanopy energy fluxes and snowpack measurements behind https://doi.org/10.1175/JHM528.1, please get in touch with me!

Thank you for the comment and sharing a reference paper with a useful dataset. We added it as a citation:

*"The forest snow model evaluations against concurrent snowpack and surface energy balance data are also surprisingly scarce (e.g. Tribbeck et al., 2006)."*